# Long-term effects of Omicron BA.2 breakthrough infection on immunity-metabolism balance: a 6-month prospective study

Yanhua Li[1,9], Shijie Qin[1,2,9], Lei Dong[3,9], Shitong Qiao[1,4,9], Xiao Wang[5,9], Dongshan Yu[6], Pengyue Gao[1], Yali Hou[7], Shouzhen Quan[3], Ying Li[3], Fengyan Fan[3], Xin Zhao [1,4,8] ✉, Yueyun Ma[3] ✉ & George Fu Gao [1,4,7] ✉

There have been reports of long coronavirus disease (long COVID) and breakthrough infections (BTIs); however, the mechanisms and pathological features of long COVID after Omicron BTIs remain unclear. Assessing long-term effects of COVID-19 and immune recovery after Omicron BTIs is crucial for understanding the disease and managing new-generation vaccines. Here, we followed up mild BA.2 BTI convalescents for six-month with routine blood tests, proteomic analysis and single-cell RNA sequencing (scRNA-seq). We found that major organs exhibited ephemeral dysfunction and recovered to normal in approximately six-month after BA.2 BTI. We also observed durable and potent levels of neutralizing antibodies against major circulating sub-variants, indicating that hybrid humoral immunity stays active. However, platelets may take longer to recover based on proteomic analyses, which also shows coagulation disorder and an imbalance between anti-pathogen immunity and metabolism six-month after BA.2 BTI. The immunity-metabolism imbalance was then confirmed with retrospective analysis of abnormal levels of hormones, low blood glucose level and coagulation profile. The long-term malfunctional coagulation and imbalance in the material metabolism and immunity may contribute to the development of long COVID and act as useful indicator for assessing recovery and the long-term impacts after Omicron sub-variant BTIs.

Over four years ago, severe acute respiratory syndrome coronavirus 2 (SARS-CoV-2) emerged, causing the coronavirus disease 2019 (COVID-19) outbreak[1]. Currently, the widespread variants of concern (VOCs) are derived from the Omicron sub-variant BA.2, including XBB, XBB.1.5,

BQ.1, BQ.1.1, BA.5.2 and BF.7. The unusually high number of mutations in the spike (S) proteins of these variants results in a sizeable antigenic shift from previous VOCs[2–4]. While BA.5.2 and BF.7 caused widespread breakthrough infections (BTIs) in China[5], XBB and XBB.1.5 infections

[1]CAS Key Laboratory of Pathogen Microbiology and Immunology, Institute of Microbiology, Chinese Academy of Sciences, 100101 Beijing, China. [2]Institute of Pediatrics, Shenzhen Children's Hospital, Shenzhen 518026, China. [3]Department of Clinical Laboratory, Air Force Medical Center, 100142 Beijing, China. [4]University of Chinese Academy of Sciences, 101408 Beijing, China. [5]School of Life Sciences, Yunnan University, Kunming 650091, China. [6]Department of Infectious Diseases, The Second Affiliated Hospital of Nanchang University, Nanchang 330008, China. [7]Shanxi Academy of Advanced Research and Innovation, Taiyuan 030032, China. [8]Beijing Life Science Academy, 102209 Beijing, China. [9]These authors contributed equally: Yanhua Li, Shijie Qin, Lei Dong, Shitong Qiao, Xiao Wang. ✉e-mail: zhaoxin@im.ac.cn; mayueyun2020@163.com; gaof@im.ac.cn

were rapidly spreading globally and accounted for approximately 90% of the total international prevalence[6]. The frequent emergence of new VOCs and the gradual weakening of the vaccine-induced immunity against the prototype strain make the current vaccine strategy inadequate in protecting against VOCs with different antigenicity. Consequently, research on vaccines, antibodies and other prophylactic measures remains challenging and seriously concerning.

Meanwhile, long coronavirus disease (long COVID or Post-COVID Conditions) has attracted overwhelming global attention. It refers to a lack of return to a usual state of health following acute COVID-19 illness, including signs, symptoms, and conditions that continue or develop after acute infection, like malfunction of major organs such as the liver, kidneys and the cardiovascular system[7,8]. Long COVID can occur in individuals regardless of vaccination status, symptom presentation, or infection with the wild-type strain, as determined primarily through questionnaires[8,9]. Omicron infections result in fewer hospitalizations, less severe illness, and a higher rate of asymptomatic cases, making it challenging to evaluate the related long COVID[10–12]. A recently published study showed that approximately 70% of Omicron BA.2 related long COVID will recover in one year after infection[13]. However, little is known about its mechanism and possible hidden pathological features.

It has been reported that the severity of long COVID is negatively correlated with vaccination status[14,15]. As countries reopen their borders and the protective effect of vaccines wanes over time, BTI numbers may rise. Moreover, the number of people immunized with vaccines alone is decreasing. Therefore, it is critical, beneficial and practical to unveil Omicron sub-variant BTIs and the associated long COVID, to understand the illness and inform future global response strategies.

Here, we first assessed the pathological characteristics and humoral immunity of individuals who had experienced a BA.2 BTI after receiving two-dose (2x) Convidecia, an adenovirus type 5 (Ad5)-vectored COVID-19 vaccine that is widely used[16–19], at three and six months post-infection (Supplementary Tables 1 and 2). We then used 4D-DIA (Data-independent acquisition) proteomics, paired single-cell transcriptomics and T cell receptor/B cell receptor (TCR/BCR) sequencing, and for an in-depth protein-to-gene level analysis and interpretation of long-term effects of COVID caused by mild BA.2 BTIs.

## Results

### Long-term effects of BA.2 BTI on major organs and blood cells
To assess long-term organ damage after a mild BA.2 BTI, we examined the clinical liver and kidney parameters after Omicron BA.2 BTIs of 2x Convidecia-vaccinated convalescents (at two time points, BA.2-BTI-3m and BA.2-BTI-6m). We observed a decrease in protein synthesis as serum total protein and albumin levels decreased three months after BTI, and then returned to normal levels six months after BTI ($p < 0.001$) (Fig. 1b). Liver enzymes cholinesterase (CHE) and α-L-fucosidase (AFU) levels were elevated three months post BTI, then decreased sharply three months later ($p < 0.01$) (Fig. 1b), indicating liver impairment at three months post BTI and recovery at six months. In particular, more individuals with elevated alanine aminotransferase (ALT) and γ-glutamyltransferase (GGT) levels were found in the BA.2-BTI-3m group, but the median with interquartile range (IQR) values were within the normal range (ALT, 0–40 U/L; GGT, 0–50 U/L) (Fig. 1b). A slightly low total bilirubin level was observed three months post BTI (Supplementary Fig. 1a). Other liver function parameters did not change significantly in the BA.2 BTI groups (Fig. 1b and Supplementary Fig. 1a).

Additionally, mild kidney injury was observed three months post BA.2 BTI, with acidic urine, low urine specific gravity, low blood urea nitrogen (BUN) levels, high serum β2-microglobulin (β2-MG) levels and fluctuating levels of serum electrolytes (potassium, calcium, chloride and phosphorus) (Fig. 1c), indicating potential impairments in

glomerular filtration, tubular reabsorption, metabolism and osmotic pressure levels of the body[20]. These changes returned to normal levels at six months post BTI (Fig. 1c). Creatinine (Cr) levels did not change significantly (Supplementary Fig. 1b). Serum magnesium ($Mg^{2+}$) levels showed extreme fluctuations (Supplementary Fig. 1b); however, the levels of $Mg^{2+}$ can be affected by many factors, especially eating. Cardiac injury was not observed in the BA.2 BTI groups (Supplementary Fig. 1c).

To assess if mild BA.2 BTIs have long-term effects on the hematopoietic system, we analyzed the levels of red blood cells (RBCs), platelets (PLTs) and white blood cells, which are critical for host immune defense, antigen clearance and wound healing (Fig. 1d). Large numbers of small RBCs were observed in the BA.2-BTI-3m group (Fig. 1d and Supplementary Fig. S1d). A lower percentage of lymphocytes (Lym%) was still evident three months post mild BA.2 BTI, but the absolute lymphocyte count was not significantly decreased (Fig. 1d). Correspondingly, the percentages of neutrophils (Neu%) and monocytes (Mono%) increased (Fig. 1d). The absolute monocyte count remained high three months after BA.2 BTI but returned to normal level after six months (Fig. 1d). We hypothesized that clearance of exogenous antigens from the body was active during the first three months after BA.2 BTI, as monocytes are well-known play important roles in antigen phagocytosis, clearance, and defense[21,22]. The absolute neutrophil count did not increase significantly, implying that the upregulation of Neu% may be caused by low Lym% (Supplementary Fig. 1d).

### Humoral immune response of patients six months after BA.2 BTI
To determine whether the convalescents could be exempt by circulating mutants and if the low Lym% affected the production of SARS-CoV-2 specific antibodies, we performed a serum pseudovirus neutralization assay against SARS-CoV and major VOCs, the prototype, Delta, Omicron sub-variants BA.1, BA.2, BA.2.12.1, BA.2.75, BA.4, BA.5, BA.5.2, BF.7, BQ.1, BQ.1.1, XBB, XBB.1.5 and CH.1.1. We found that hybrid immunization with 2x Convidecia vaccination and BA.2 BTI generated durable and broadly neutralizing antibodies (NAbs) against the prototype, Delta, BA.1, BA.2, BA.2.12.1, BA.2.75, BA.4/5, BF.7, BQ.1 and BQ.1.1 sub-variants, and moderate cross-neutralization against SARS-CoV for at least six months (Fig. 2). However, there was limited neutralization against the XBB, XBB.1.5 and CH.1.1 sub-variants (Fig. 2i–k and Supplementary Fig. 2). All results of the BTI group were significantly higher than those of the control group ($p < 0.0001$) (Fig. 2, Supplementary Fig. 2). This indicates that the humoral immune response remains active six months after BTI and maintains a certain preventive effect against some potential VOCs infections.

### Proteome analysis supports coagulation disorder and immunity-metabolism imbalance
Next, to determine the underlying factors driving long-term impacts, we used proteomic analysis to comprehensively characterize changes in the serum proteome of convalescents six months after BA.2 BTI. A total of 1824 proteins were identified in 18 samples (including 10 mild BA.2 BTI convalescents and eight healthy persons, and all samples were from male participants) in this study. Our results showed that the protein expression profiles of the BA.2-BTI-6m and control groups were divided into two clusters (Fig. 3a). Differential analysis identified 22 significantly upregulated and 58 significantly downregulated proteins compared with the controls in the BA.2-BTI-6m group (Fig. 3b, c and Supplementary Data 1). The main upregulated proteins in the BTI group were coded by genes *IGKV1D-33, UNC13D, CARHSP1, SYTL4, LIMS1* and *ARHGAP45*, while the downregulated included *MT-CO2, ATP5MG, ALDH7A1, PLCXD1* and *SULT2B1* (Fig. 3b). In particular, *PSMD12, PHGDH, DNASE1L2* and *RDH12* were detected only in the control group (Fig. 3b). Functional analysis showed that the

upregulated proteins were mainly involved in actomyosin structure, wound healing, PLT, mononuclear cell migration, cell adhesion, and GTPase activity, suggesting that the body was still in a state of stress and immune activation six months after mild BA.2 BTI (Fig. 3d). The downregulated proteins were mainly involved in ribonucleotide, nucleotide metabolism, ATP synthesis, mitochondrial metabolism and proton transmembrane transport, indicating that the material metabolism and oxidative energy supply of recovering patients were affected in the long term (Fig. 3d). Specifically, some multifunctional proteins in the BA.2-BTI-6m group, such as *F11R, CSRP1, LIMS1, PPKAR1A, CD47* and *SYTL14* genes coded proteins, jointly maintained active biological signals (Fig. 3e). In contrast, the downregulated key proteins coded by *ATP5MG, ATP5F1C, ATP5PD, ATP5PO, COX2, DLD, DLST, ENO3, PGK2, PGM1* and *FASN* collectively downregulated multiple metabolism-related pathways, including ATP, nucleotide and protein metabolism (Fig. 3f). Additionally, from the distribution of the variable proteins ($p < 0.05$) in the Kyoto Encyclopedia of Genes and Genomes (KEGG) Level 2 diagram (Supplementary Fig. 3a), the overall immune and cellular immune status of the BA.2-BTI-6m group was observed to be more active. There were more upregulated variable

proteins related to the immune system, especially those involved in immunity against foreign antigens (viruses, bacteria and parasites), than downregulated ones. However, more variable proteins in metabolism-related pathways, such as nucleotide, vitamin, lipid, glycan, energy, carbohydrate and amino acid metabolism, were downregulated. In particular, the protein-protein interaction analysis showed that the jointing proteins, coded by *CD47, MFGE8, EPX, F11R, TFPI, ILK, ITGA2B, FCER1G, PHGDH, LMNA* and *FASN* were involved in two or more pathways related to immunity, coagulation and metabolism, indicating key roles in regulating these pathways (Supplementary Fig. 3b). In addition, we conducted gene set enrichment analysis (GSEA) on the liver, lung, pancreas, heart specific protein gene sets, and coagulation system (Supplementary Fig. 4a), which may reflect potential tissue damage[23,24]. The results showed that coagulation in the BA.2-BTI-6m group was significantly activated ($P = 0.0012$), with the expression of DEG and coagulation intersection genes *CSRP1, ILK, ITGA2B, F11R, SELP, FCER1G,* and *TFPI* upregulated (Supplementary Fig. 4b). The GSEAs of the liver, lungs, pancreas, or heart were insignificant (Supplementary Fig. 4a, b), indicating no considerable organ

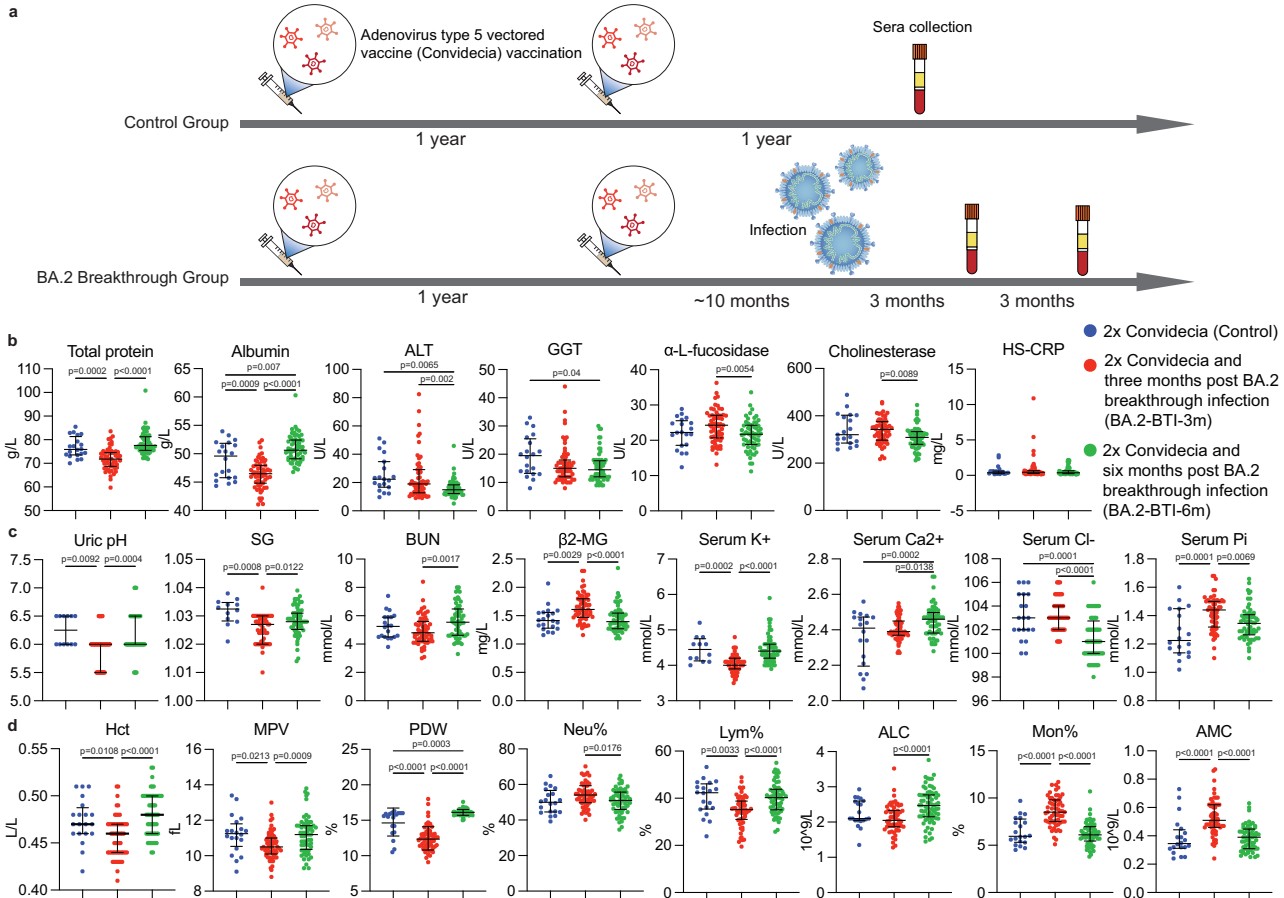

**Fig. 1 | Study design and results of routine blood examinations from convalescents of Omicron BA.2 breakthrough infection (BTI). a** Schematic diagram of the study design. **b** Changes in liver function parameters three months post BA.2 BTI ($n = 59$, one participant missed this time of routine blood test) and six months post BA.2 BTI ($n = 60$), ($n = 20$ biologically independent samples in the Control group, one missing value in ALT, Cholinesterase and HS-CRP, separately). **c** Varied parameters of renal function and electrocytes ($n = 20$ biologically independent samples in Control group, eight persons did not take the tests of Uric pH, SG and serum K+, two missing values in Ca2+ and Pi, separately; $n = 59$ for BA.2-BTI-3m; $n = 60$ for BA.2-BTI-6m). **d** Dynamic changes in major red and white blood cell levels, three months after BA.2 BTI ($n = 59$) and six months after BA.2 BTI ($n = 60$),

($n = 20$ biological replicates in the Control group). The Control group is colored in blue. Groups of BA.2-BTI-3m and BA.2-BTI-6m were stratified by time after BA.2 BTI on 2x Convidecia, and colored in red and green separately. ALT alanine aminotransferase, GGT γ-glutamyltransferase, HS-CRP hypersensitive C-reactive protein, SG urine specific gravity, BUN blood urea nitrogen, β2-MG serum β2-microglobulin, K+ potassium, Ca2+ calcium, Cl- chloride, Pi phosphorus, Hct Hematocrit, MPV mean platelet volume, PDW platelet distribution width, Neu neutrophil, Lym lymphocyte, ALC absolute lymphocyte count, Mono monocyte, AMC absolute monocyte count. Convidecia Ad5-vectored COVID-19 vaccine. Data are presented as median with interquartile range. *P* values reflect two-sided ordinary ANOVA tests adjusted for multiple comparisons. Source data are provided as a Source Data file.

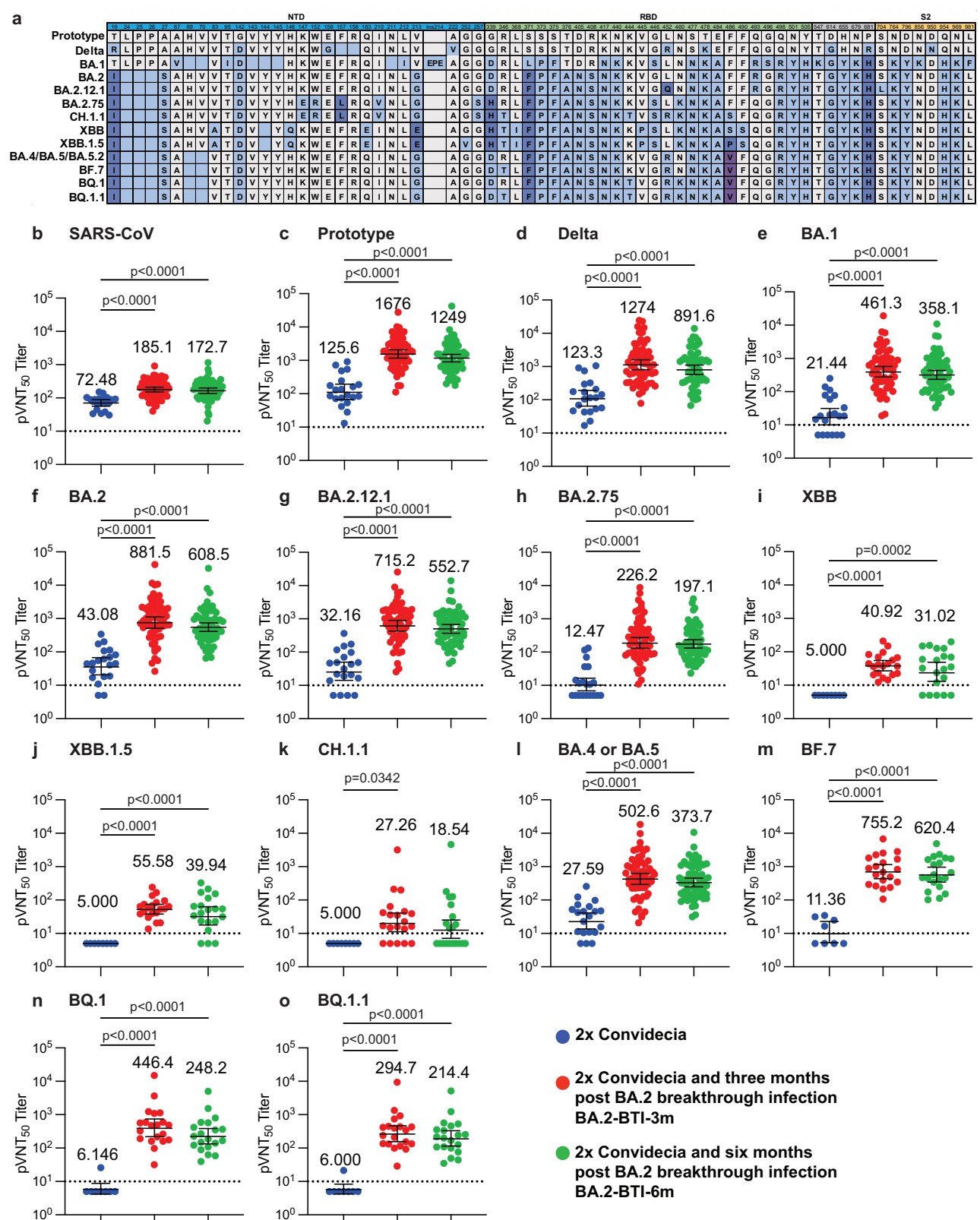

damage was found at six months post BA.2 BTI, consistent with clinical results.

### Single-cell compositional characteristics of PBMC from BA.2-BTI patients

To further characterize the recovery of patients six months post BA.2-BTI, we used single-cell RNA sequencing (scRNA-seq) to assess changes in peripheral blood mononuclear cell (PBMC) composition, gene expression and immune repertoire (all scRNA-seq samples were from male participants). After quality control, we obtained nine samples of single-cell transcriptome data, including 45,532 cells in the control group (four participants) and 54,653 cells in the BA.2-BTI-6m group (five mild BA.2 BTI convalescents) (Fig. 4a). Combined with Azimuth prediction algorithm and artificial correction, we identified 18 cell

**Fig. 2 | Neutralizing antibody levels of SARS-CoV-2 convalescents' sera against the pseudoviruses of SARS-CoV-2 variants. a** The mutations on the spike protein of SARS-CoV-2 variants. The amino acid differences from prototype (including substitution, deletion, and insertion) in sub-variant sequences are highlighted in blueish color. Deletion mutations are shown as blank squares with background color. **b–o** 50% pseudovirus neutralization titers (pVNT$_{50}$) against SARS-CoV, SARS-CoV-2 PT, Delta, and Omicron BA.1, BA.2, BA.2.12.1, BA.2.75, BA.4/BA.5/BA.5.2 ($n = 20$ biologically independent samples in the Control group; $n = 60$ for BA.2-BTI-3m; $n = 60$ for BA.2-BTI-6m), XBB, XBB.1.5, CH.1.1, BF.7, BQ.1, and BQ.1.1 (for the last six sub-variants, $n = 8$ for Control; $n = 20$ for BA.2-BTI-3m; $n = 20$ for BA.2-BTI-6m). The Control group is colored in blue. Groups of BA.2-BTI-3m and BA.2-BTI-6m were stratified by time after BA.2 BTI on 2x Convidecia, and colored in red and green separately. The pVNT$_{50}$ in each group is shown as the geometric mean titers (GMTs) with 95% confidence intervals (CI). GMT numbers are shown on top of each column. The lowest limit of the pseudovirus neutralization assay (1:10) is shown by dashed lines. A pVNT$_{50}$ < 10 indicated a NAb-negative sample and was counted as 5. $P$ values reflect two-sided ordinary ANOVA tests adjusted for multiple comparisons. Source data are provided as a Source Data file.

types, including CD4 T cell subclusters, CD8 T cell subclusters, γδ T cells (gdT), Mucosal-associated invariant T (MAIT) cells, B cells, natural killer (NK) cells, monocyte (Mono) subclusters, dendritic cells (DC) plasmacytoid dendritic cells (pDC) and PLTs (Fig. 4a, c). These cell types were evenly distributed between the control and BA.2-BTI-6m groups without significant between-batch effects (Fig. 4b and Supplementary Fig. 5). Furthermore, we used a more refined Milo algorithm to identify potential differences in cell cluster abundance and found no significant differences in the ratio of multiple cells between the control and BA.2-BTI-6m groups (Fig. 4d–f). This indicates that the cell ratio returned to normal levels, consistent with the results of the clinical laboratory tests. We also calculated the DEGs for the different cell types and found that NK cells had the highest number of DEGs, indicating that their gene expression was affected in the long term (Fig. 4g). Additionally, we examined the functional modules of DEG enrichment in the different cell types and found that the activation modules shared by PBMCs were concentrated in response to unfolded or topologically incorrect protein, transcription, leukocyte/T cell activation, and antigen presentation, intrinsic apoptotic signaling, etc. (Fig. 4h). In contrast, the shared downregulation module focused on cytoplasmic translation, ribosome processing and assembly, ATP metabolic process, and protein ubiquitination regulation, etc. (Fig. 4i).

### Innate immune cells were profoundly affected by BA.2 BTI
Because the proportion of innate immune cells in the BA.2-BTI-6m group was similar to that in the control group, we investigated gene expression changes in these cells to further reveal any deeper and hidden long-term influences. We first focused on the NK and CD14 Mono cells, which exhibited more DEGs (Fig. 4g). We first examined the changes in NK cell effector proteins such as granzyme and perforin. The results showed that GZMH and GNLY were significantly upregulated in the BA.2-BTI-6m group, while GZMB, PRF1, and IFNG were significantly downregulated. However, all logFCs were between 0.25 and 0.5, indicating NK cells had become mildly active after six months of recovery from infection. Subsequent NK cell DEG analysis showed that NK cells had 305 DEGs (Supplementary Data 2), and the most upregulated genes were *TUBB4B, MTRNR2L12, FOSB, TUBA1B*, and *KLF2*, etc. while the most downregulated genes were *XCL1, IFITM3, LAIR2, CCL4* and *MYOM2*, etc. (Fig. 5c). Functional analysis indicated the enrichment of related processes such as T cell activation, response to virus, leukocyte adhesion and leukocyte immunity and interferon, etc. (Fig. 5d). These results showed that the NK cells of the BA.2-BTI-6m group maintained stronger immune activation than those of the control group. In addition, there were 266 DEGs in CD14 Mono (Supplementary Data 2), and we further clustered CD4 Mono into four subpopulations and tried to identify MS1 related subpopulations[25,26], which have been reported to be associated with severe COVID-19 and sepsis, and their changes at six months after infection are unknown. The results suggested that cluster 0 is potential MS1, with high expression of MS1 markers such as *ALOX5AP, RETN, THBS1*, etc[25,26] (Fig. 5e, f). In addition, this cluster also highly expressed S100A8 and S100A9, which are highly expressed in MS1 compared to MS2 and MS4[25,26] (Supplementary Data 3). We also calculated the scores of the top 50 highly expressed genes in MS1[25,26] and found that they were significantly enriched in cluster 0. Further analysis identified 202 DEGs

in MS1 (Supplementary Data 4). The top genes upregulated in the BA.2-BTI-6m group included *TMEM176B, TMEM176A, HLA-DQA1, CD52, DNAAF1*, etc., while the downregulated genes were *TMA7, AC020656.1, SMDT1, DDIT4*, etc. (Fig. 5g). Functional enrichment analysis showed that MS1 related cells in the BA.2-BTI-6m group exhibited active antigen presentation, T cell activation, and hematopoietic regulation, while ribosomal biogenesis and ATP metabolism processes were inhibited (Fig. 5h). The results of the gene set scoring were consistent with the enrichment analysis, that the myeloid monocyte exhibited increased antigen-presenting activity and reduced protein processing activity (Fig. 5i). These results indicate that basic protein synthesis in the myeloid cells in the BA.2-BTI-6m group decreased, but still retained a good immune pre-activation state.

### T cell expression and TCR repertoire six months after BA.2 BTI
We further analyzed the differences in cytotoxic responses among the nine T cell subsets (Fig. 6a), finding the cytotoxicity of the T cell subsets of the BA.2-BTI-6m group was heterogeneous. Specifically, the T cell cytotoxicity of effector memory CD8 T cells (CD8 TEM) in the BA.2-BTI-6m group was significantly increased, while that of gdT cells were significantly reduced, suggesting that the impact of BA.2 BTI on them was more persistent comparing with other subsets (Fig. 6b). We also focused on Treg cells, which play important roles in maintaining immune suppression and tolerance during infection. The results showed that there was no significant difference in Treg activation score or DEG (Fig. 6c), but the IRF1 gene related to Treg activity inhibition was significantly upregulated in the BA.2-BTI-6m group (Fig. 6d). Further differential analysis identified 51 DEGs of Treg, and functional enrichment showed that they were mainly involved in RNA splicing, response to unfolded proteins, regulation of protein ubiquitination, and phosphatase activity (Fig. 6e).

We simultaneously examined the TCR sequences to clarify the T cell repertoire changes. Overall, except for gdT cells, for which fewer TCR sequences were detected, T cells detected nearly 50%-60% of the TCRs (Fig. 6f, g). TCR clonal diversity analysis showed no significant difference between the BA.2-BTI-6m and control groups, indicating that the TCR repertoire of BA.2-BTI-6m had stabilized at this time point (Fig. 6h, i). Further analysis showed that clonal expansion differed between the different T cell subsets and the significantly amplified clonotypes were mainly concentrated in the CD8 TEM and effector memory CD4 T cells (CD4 TEM), both in the control and BA.2-BTI-6m groups (Fig. 6j). This suggests that memory T cell levels might undergo a lasting dynamic shift after vaccination or BTI. Additionally, analysis showed the moderately amplified TCR in the BA.2-BTI-6m group was 2.637% higher than that in the control group, while the difference in other types of TCR was about 1% (Fig. 6k). The presence of CDR3 sequences in the control group may reflect a vaccine-induced clonal type, while CDR3 clonotypes from infected group may mediate the T cell re-response to the virus. We respectively extracted 10 specific most-amplified CDR3 clonotype sequences from the control and BA.2-BTI-6m groups (Fig. 6l). As expected, there were significant differences in TCR and limited proportion of shared clones between individuals (Fig. 6m, n). Considering that the expanded T cells mainly concentrate on CD8 TEM, we selected CD8 TEM to further study the clonality-associated differences. Compared with non-expanded T cells, a total of

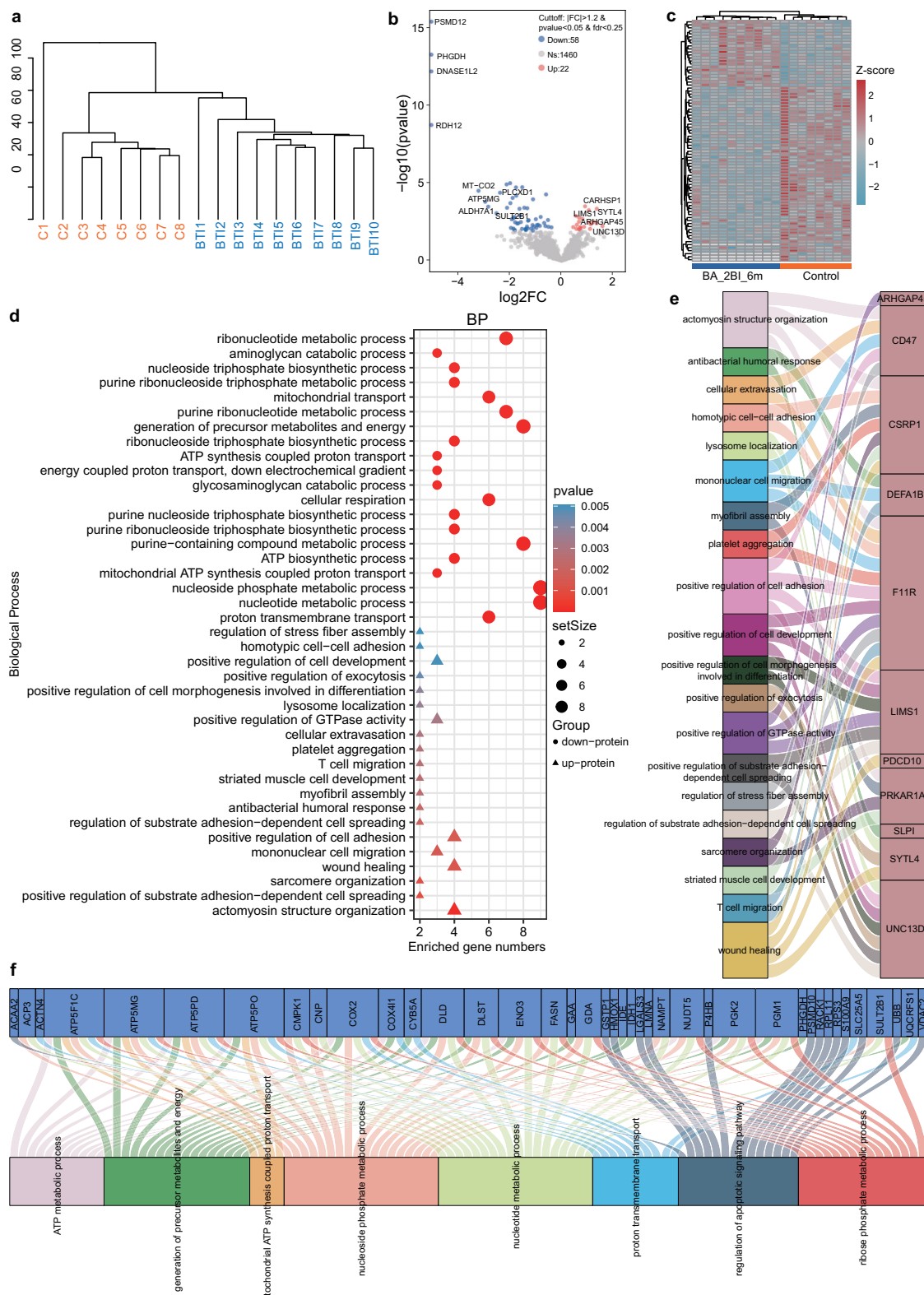

Fig. 3 | Proteomic analysis of differences in peripheral blood mononuclear cells between BA.2-breakthrough infection (BTI)-6m and control groups. a Tree cluster diagram depicts sample separation between the two groups. b Volcanic map indicates differential proteins between the two groups. c The heat map shows the differential proteins in the control and BA.2-BTI-6m groups. d Bubble chart displays the biological processes and functions of the enriched differentially expressed proteins. e Sankey diagram shows relationships between upregulated proteins and enriched biological processes. f Sankey diagram shows relationships between downregulated proteins and enriched biological processes.

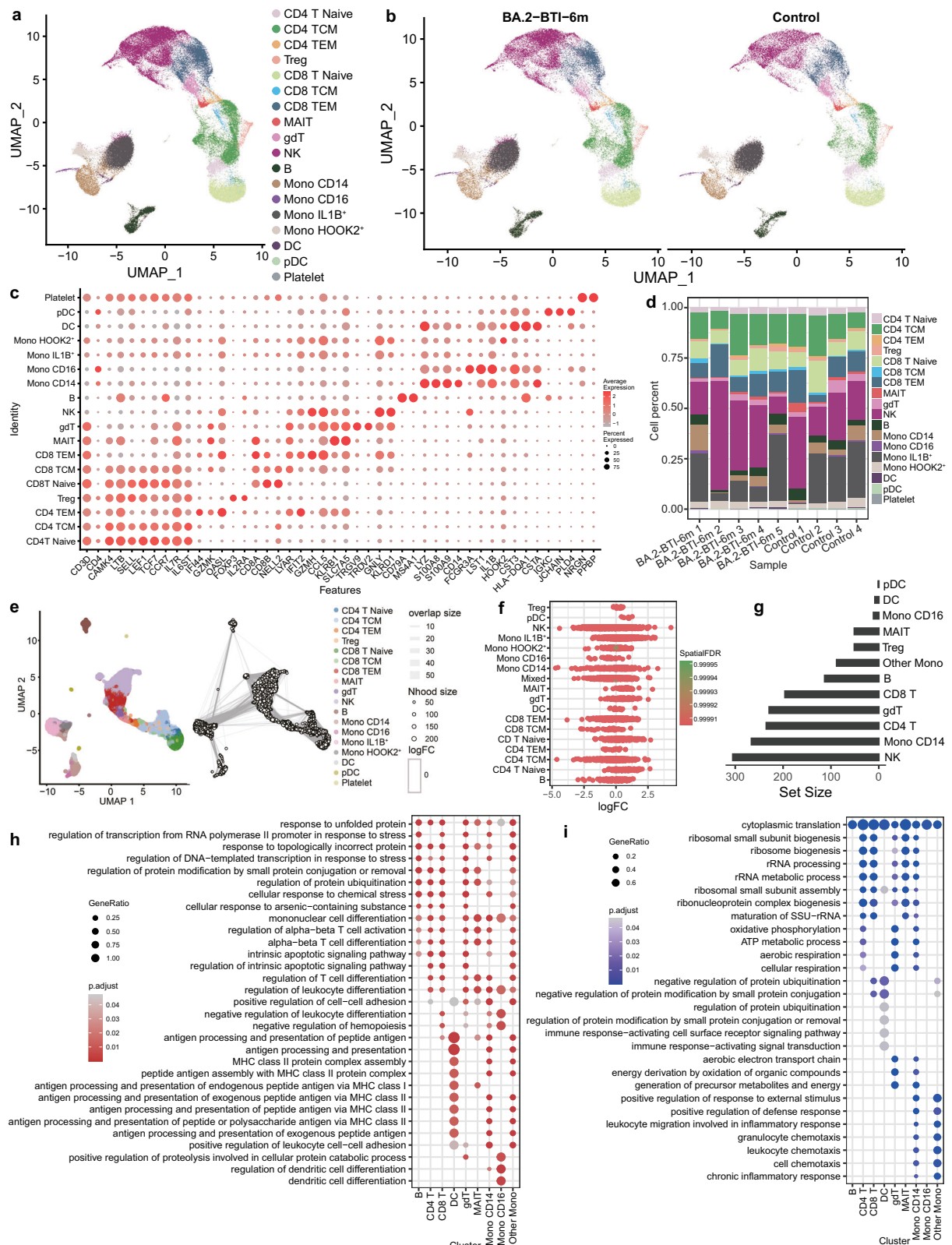

132 DEGs were found (Supplementary DatV), and the top 10 significantly upregulated genes in expanded T cells were *GZMB, GNLY, LGALS1, FGFBP2, KLRD1, PRF1, S100A6, S100A4, GZMH* and *ADGRG1*, while the top 10 downregulated genes were *GZMK, LTB, DNAJB1, CMC1, IL7R, TCF7, DUSP2, HSPA1B, COTL1* and *RGCC*. Clearly, expanded T cells upregulated cell killing effector proteins such as granzyme (*GZMB, GZMH*), perforin (*PRF1*) and granulysin (*GNLY*), and downregulated

naive T cell markers such as *LTB, IL7R* and *TCF7*, etc (Fig. 6o). Further functional enrichment analysis showed that upregulated genes were mainly involved in T cell activation, T cell differentiation, and host defense, while downregulated genes were not only enriched in T cells, but also involved in hematopoietic regulation, calcium ion response, and response to topologically incorrect proteins and unfolded proteins (Fig. 6p). These results suggested that differences were found in

**Fig. 4 | Single-cell transcriptome sequencing characterized cellular and molecular changes of peripheral blood mononuclear cells (PBMCs). a** UMAP graph displays the cell types and clustering of PBMCs. Firstly, the Azimuth algorithm is used to map the data to the reference cell set of PBMC, and then combined with specific high expression genes to manually determine the cell type. **b** UMAP graph shows cell types and clustering of PBMCs in control and BA.2-BTI-6m groups. **c** Dotplot diagram with marker genes corresponding to different cell types. Naïve CD4 T cells (CD4T Naive) (CD3D/CD4/CAMK4/LTB/SELL/TCF7/CCR7), Central memory CD4 T cells (CD4 TCM) (CD3D/CD4/CAMK4/IL7R/IL6ST), Effector memory CD4 T cells (CD4 TEM) (CD3D/CD4/CAMK4/IIFI44/GZMK/OASL), Regulatory T cells (Treg) (CD3D/CD4/CAMK4/FOXP3/IL2RA), Naïve CD8 T cells (CD8T Naive) (CD3D/CD8A/CD8B/LTB/SELL/TCF7/CCR7), Central memory CD8 T cells (CD8 TCM) (CD3D/CD8A/CD8B/NELL2/LYAR), Effector memory CD8 T cells (CD8 TEM) (CD3D/CD8A/CD8B/ IFIT2/GZMH/CCL5), Mucosal-associated invariant T (MAIT) cells (KLRB1/SLC7A5), γδ T cells (gdT) (TRGV9/TRDV2), natural killer (NK) cells (GNLY'/ KLRD1), B cells (CD79A/MS4A1), CD14 monocyte (LYZ/S100A8/S100A9/CD14), CD16 monocyte (LYZ/FCGR3A/LST1), IL1B$^+$ monocyte, HOOK2$^+$ monocyte, dendritic (DC) cells (CST3/HLA-DQA1/CSTA), plasmacytoid dendritic (pDC) cells (IGKC/JCHAIN/PLD4) and platelet (PLTs) (NRGN/PPBP). **d** Stacked bar charts display the proportion of different cell types in a single sample. **e** UMAP plot shows differences in cell composition between control and BA.2-BTI-6m groups identified based on Milo algorithm. **f** The figure shows the fold change of cell composition in the control and BA.2-BTI-6m groups identified based on Milo algorithm. **g** Number of differentially expressed genes in different cell types in the control and BA.2-BTI-6m groups. **h** Bubble chart displaying biological functions enriched by significantly upregulated genes in different cell types. **i** Bubble chart displaying biological functions enriched by significantly downregulated genes in different cell types. *P* value is calculated through hypergeometric testing and adjusted by the Benjamin and Hochberg methods.

the functions of expanded and non-expanded T cells six months after BTI. Finally, we compared VDJ gene frequency between the BTI and control groups. The results showed no significant difference in the frequency of most V genes between the TRA and TRB chains in the BA.2-BTI-6m and control groups, except for the low-frequency-specific TRAV18 gene and the significantly downregulated TRBV6-6 gene ($p = 0.016$) (Supplementary Fig. 6a, b) detected in the BA.2-BTI-6m group. Moreover, the frequency of TRAJ29 and TRAJ21 of the TRA chain in the BA.2-BTI-6m group was significantly reduced ($p = 0.016$ and $p = 0.032$, respectively) (Supplementary Fig. 6c), as was the frequency of TRBJ1-4 ($p = 0.016$) of the TRB chain (Supplementary Fig. 6d). Conversely, the frequency of TRAJ45 and TRAJ37 genes in the BA.2-BTI-6m group significantly increased ($p = 0.016$ and $p = 0.032$, respectively) (Supplementary Fig. 6c), possibly related to specific T cells being enriched against BA.2 variants. Overall, although the proportion of T cells and TCR diversity returned to control levels six months post BTI, the T cells maintained some antiviral activity.

## B cell expression and BCR repertoire six months after BA.2 BTI

Here, we subdivided the B cell subpopulation and identified four main subsets: naïve B cells (MS4A1 + IGHD + ), germinal center B cells (MS4A1 + NEIL1 + ), memory B cells (MS4A1 + CD27 + ) and intermediate B cells (IGHD + CD27 + ) (Fig. 7a, b). Activity analysis showed that the BCR signals and B cell activation scores of Naive B, memory B, and intermediate B cells still significantly increased six months after infection (Fig. 7c). Differential analysis showed that the number of DEGs in the B cell subpopulations was not high, with a maximum of 65 in Naive B cells (Supplementary Data 6 and Supplementary Fig. 7a), indicating that the expression status of B cells at this time had approached to the level of the control group. Specifically, the genes significantly upregulated in Naive B cells of the BA.2-BTI-6m group include *HSPA1B, DNAJB1, FOSB, HSP90AA1, IFRD1, HSPA8, PRDX1, HSPH1, FOS*, etc. The downregulated genes include *RPS26, TMA7, SMDT1, CRIP1, ARHGAP24, RPL37, RPS29, ATP5F1E, RPS21*, and *RPL37A*, etc. (Fig. 7d, e). KEGG enrichment analysis showed that downregulated genes mainly participated in the ribosome and COVID-19 pathways, while upregulated genes were involved in antigen presentation, IL-17 signaling pathway, MAPK signaling pathway, and infection pathways (Fig. 7f). Next, we explored the correlation between neutralizing antibody levels and key signals in BA.2-BTI-6m subjects. Overall, in addition to the JAK-STAT signaling pathway, antigen presentation, interferon, B cell activation and BCR signaling were positively correlated with the neutralizing activity of various SARS-CoV-2 variants (Fig. 7g). However, due to the small sample size, these positively correlated trends are not significant and therefore worth of further validation.

We also examined the BCR sequence and found the B cell subpopulations with paired heavy and light chains accounted for approximately 40–65%. (Fig. 7h, i). The BCR repertoire diversity analysis showed no significant differences between the BA.2-BTI-6m and control groups (Fig. 7j and Supplementary Fig. 7b), indicating that the infected B cells had returned to normal status after six months. Unlike the T cells, we did not find medium- or high-frequency clono-B cells in the BCR pool, but only some low-frequency amplified clonotypes (Fig. 7k). A slightly higher proportion of amplified BCR was observed in BA.2-BTI-6m compared to the control group (3.258% vs 2.336%) (Fig. 7k), indicating that the B cells in the BA.2-BTI-6m group retained a slightly more active status, possibly related to the BA.2-induced long-term immune activation, consistent with the serum neutralization results. Due to the lack of enriched B cells and high heterogeneity, we did not detect BCR intersections between individuals (Supplementary Fig. 7c, d). Further analysis revealed that the detected clonal expansion was mainly concentrated in the intermediate B, memory and naïve cell clusters (Fig. 7l). Considering that this BCR clonotype may be the source of broad-spectrum NAbs against SARS-CoV-2, we extracted the CDR3 sequence of the specific clonal amplification BCR from both the BTI and control groups (Fig. 7m). Among them, CARDRGYNTEGF-DYW_CAAWDDSLSGHVF was the specific clonal type with the highest proportion in the infected group. Finally, we analyzed the use of infection-induced VDJ in the BA.2 strain. We found that the frequency of IGLV3-1 for BCR light chains and IGHV4-61 for BCR heavy chains increased significantly in the infected group ($p = 0.029$) (Supplementary Fig. 7e–h), which may be specific VDJ rearrangements in response to BA.2. In contrast, the heavy-chain BCR of the control group used IGHV1-3 more frequently ($p = 0.036$) (Supplementary Fig. 7e–h). No significant differences were detected in the frequency of the J gene fragment (Supplementary Fig. 7e–h). Overall, the B cells maintained stronger antiviral activity and more specific clonal expansion, although the B cell proportions and BCR diversity returned to control levels six months after the BTI.

## Changes in parameters related to metabolism profile of patients after BA.2 BTI

To confirm the hypothesis we obtained from proteomics and transcriptomics, we retrospectively analyzed other clinical parameters related to immunity and metabolism. In particular, hormone disorder was observed after BA.2 BTI, as prolactin, estradiol, T3, T4 and cortisol levels were upregulated three months post BA.2 BTI, and prolactin and cortisol levels did not return to normal even after six months (Fig. 8a). The coordination of various hormones maintains the metabolism and function of the body[27,28]. Prolactin has a bioactive function acting as a hormone and a cytokine, and can regulate cellular immunity and humoral immunity through the expression of prolactin receptors on immune cells, such as T cells, B cells and NK cells[28]. But prolactin may be harmful if overactive[28]. Cortisol plays a crucial role in the body's substance metabolism, homeostatic stress response, immune function and physiological functions of various organs[29,30]. Cortisol also plays a key role in regulating stress responses, and hypocortisolism has substantial clinical overlap with COVID-19 symptoms. But cortisol may be

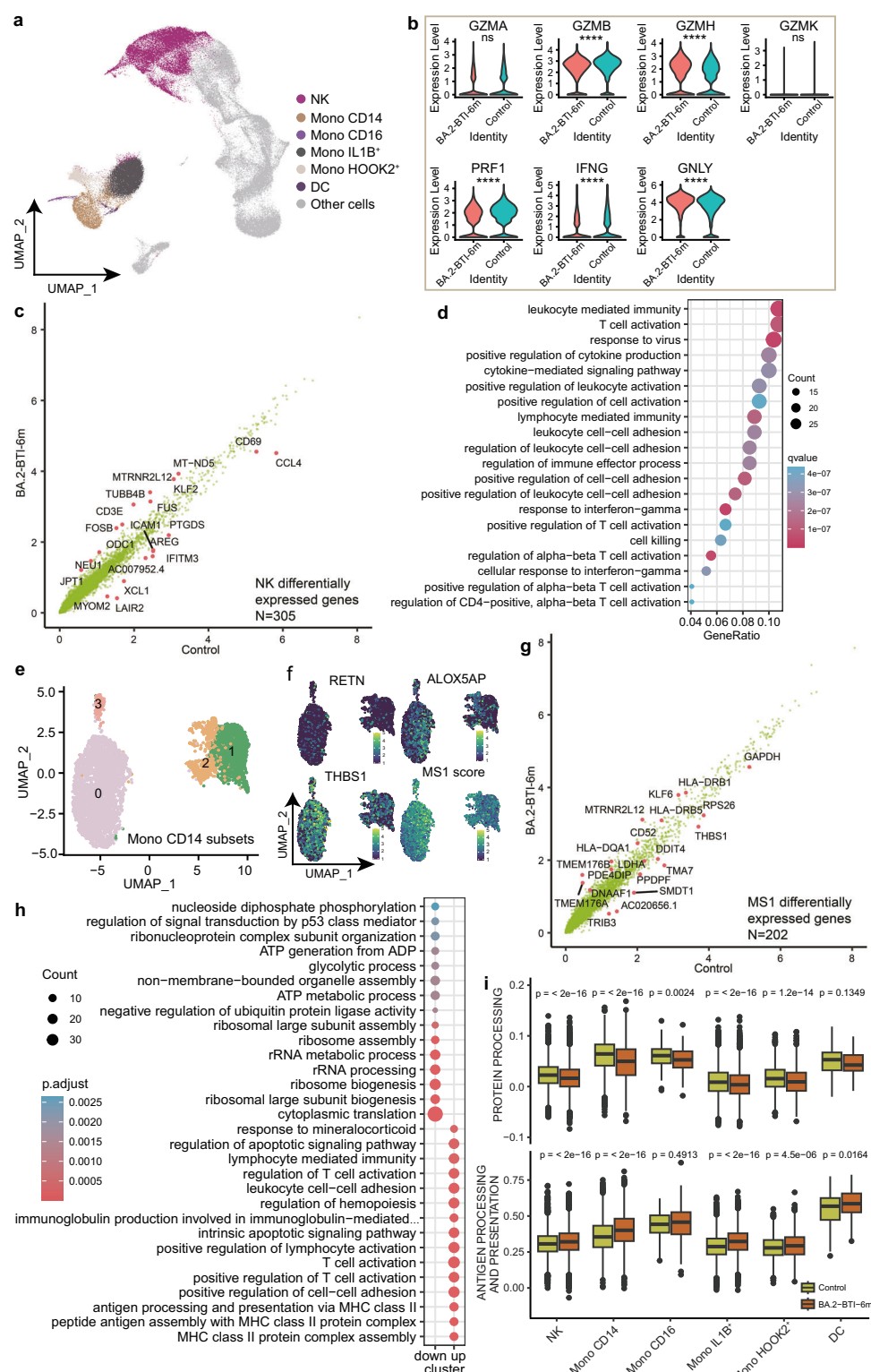

**Fig. 5 | Single-cell transcriptome sequencing was used to characterize cellular and molecular changes in innate immune cells. a** UMAP graph visualizes type and clustering of innate immune cells. **b** Violin diagram showing the expression level of NK cell effector proteins in the control and BA.2-BTI-6m groups. *P* values are calculated by the two tailed Wilcoxon rank sum test, ****$p < 0.0001$, ns $p > 0.05$. **c** Volcanic map showing differentially expressed genes in NK cells in both groups. **d** Bubble chart displaying biological functions enriched by differentially expressed genes in NK cells. **e** UMAP plot showing subpopulations of CD14 monocytes. **f** UMAP plot showing the expression level of MS1 cell gene marker. **g** Volcanic map showing differentially expressed genes in MS1 cells in both groups. **h** Bubble chart displaying biological functions enriched by differentially expressed genes in MS1 cells. *P* value is calculated through hypergeometric testing and adjusted by the Benjamin and Hochberg methods. **i** Box plot showing differences in protein processing, and antigen-presenting activity scores of innate immune cells. Minima: Lower limit of the whisker; Maxima: Upper limit of the whisker; Centre: Median line inside the box; The upper and lower box bounds represent the 25% and 75% percentile of data. Two tailed Wilcoxon rank sum test.

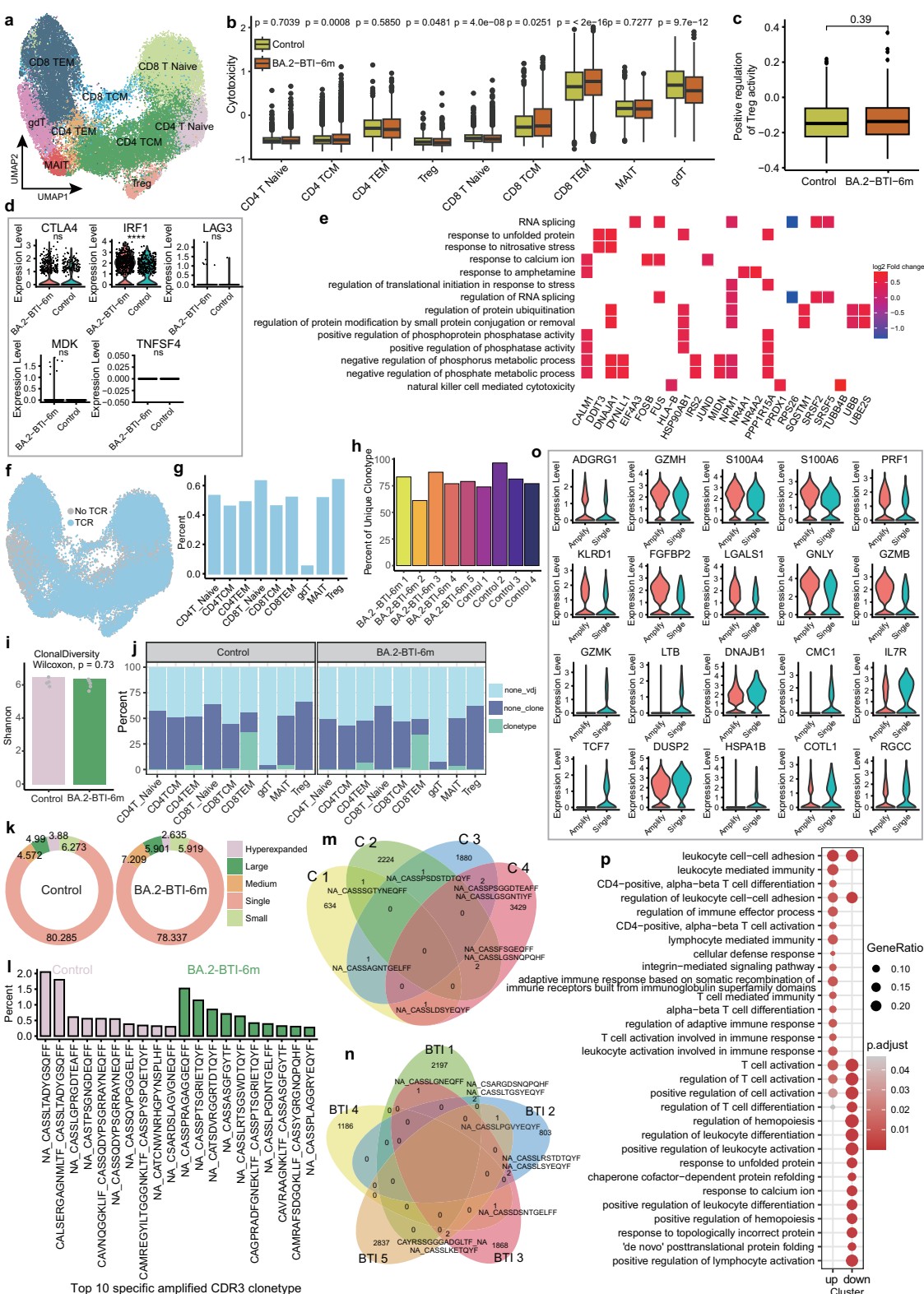

higher during moderate-intensity stress[31]. Research showed that post-COVID-19 unemployment rate increases during the epidemic[32,33], and participants' concern about their own recovery after infection and the stress of life and work may induce an increase in cortisol. In addition, a low level of cortisol was listed as an independent predictive biomarker of long COVID after PT infection[34], whether it is also applicable to the development of COVID-19 after Omicron BTI remains to be studied. Cortisol has an immunosuppressive effect. It treats conditions caused

by an overactive B cell-mediated antibody response[29,30]. The increase in cortisol seen at six months of infection may be a protective mechanism against the over activation of the immune response.

Importantly, liver and kidney injury and abnormal hormone levels can induce metabolic disorders, as indicated by the extreme changes in the mid-term serum glucose level (glycated albumin, GA), which did not recover to the control levels even six months after BA.2 BTI (Fig. 8a). This indicated that glucose metabolism was affected in the

**Fig. 6 | Single-cell sequencing characterized changes in T cell molecules and TCR. a** UMAP graph displaying T cell subpopulations and clustering. **b** Box plot showing difference in T cell toxicity scores between both groups. **c** Box plot showing difference in Treg activation gene set scoring between both groups. Minima: Lower limit of the whisker; Maxima: Upper limit of the whisker; Centre: Median line inside the box; The upper and lower box bounds represent the 25% and 75% percentile of data for (**b**) and (**c**). Two tailed Wilcoxon rank sum test. **d** Violin diagram showing the expression level of Treg activity inhibitory gene in the control and BA.2-BTI-6m groups. *P* values are calculated by the two tailed Wilcoxon rank sum test, ****: *p* < 0.0001, ns: *p* > 0.05. **e** The heat map shows the biological process of Treg differential gene enrichment. **f** UMAP diagram showing T cells with TCR information. **g** Bar chart indicating proportion of TCR detected in T cell subsets.

**h** Bar chart showing TCR clonal proportion of a single sample. **i** Bar chart showing difference in TCR diversity between both groups. *P* values are calculated by the two tailed Wilcoxon rank sum test, Control (*n* = 4), BA.2-BTI-6m (*n* = 5). **j** Stacked bar chart displaying proportion of clonal T cell subpopulations. **k** The circular plot shows the proportion of TCR with different frequencies in the control and BA.2-BTI-6m groups. **l** Bar chart showing top 10 TCR CDR3 sequences specifically amplified in control or BA.2-BTI-6m groups. **m** Venn chart showing intersection of TCR in control group samples. **n** Venn chart showing intersection of TCR of samples from BA.2-BTI-6m group. **o** The violin diagram shows the differential genes upregulated and downregulated by top 10 in CD8 TEM. **p** Functional enrichment analysis of upregulated and downregulated DEGs by CD8 TEM.

long run. Cortisol promotes the activation of glycogen phosphorylase, which is required for the effect of adrenaline on glycogenolysis[29,30]. Therefore, changes in cortisol may correlate with both the immunity and the abnormal GA. Lipid metabolism was also disturbed, with the levels of lipase and low-density lipoprotein (LDL) first decreased and then increased at three and six months post BTI, respectively (Fig. 8a) but with recovery trends.

### Long-term impacts of BA.2 BTI on coagulation profile

Coagulation profile was found impaired with routine blood testing and proteomic analysis. Critical impairments were further proved by coagulation testing, as indicated by a reduction in prothrombin time activity (PTA) and fibrinogen (FIB) level, increase in prothrombin time (PT), PT ratio (PTR), thrombin time (TT) and international normalized ratio (INR) three months after mild BA.2 BTI (Fig. 8b). In particular, PT, PTR, activated partial thromboplastin time (APTT), APTT ratio and INR further increased along with a further reduction in PTA six months post BA.2 BTI. TT and FIB levels recovered to normal levels six months post BA.2 BTI (Fig. 8b), indicating that the coagulation profile recovery may take longer. Well, our finding was inconsistent with previous reports on hypercoagulability caused by COVID-19[35–37]. It could be because of the stage and the severity of the disease, coagulation substances were consumed in large quantities at first during the infection period (hypercoagulability state) and with the disease progression, entered a state of deficiency of coagulation substances later. The principle is similar to the occurrence of disseminated intravascular coagulation (DIC)[38].

## Discussion

Current COVID-19 vaccines are not effective against all BTIs in distant variants. Moreover, public anxiety has been compounded by the persistence of long COVID. Vaccination or infection with a milder variant may mitigate the symptoms of long COVID[39–41], but the underlying long-term health impacts and mechanism remain unclear. Our report comprehensively analyzed long COVID or long-term effects of COVID resulting from a mild BA.2 BTI, we examined the progression from the protein level to the single-cell transcriptome level and confirmed with clinical testing.

Long COVID is related to severe SARS-CoV-2 variant infections, especially the prototype isolate, and may last longer than a year[39,40]. Previously, we reported that severe systemic tissue damage was found in patients with moderate and fatal prototype-related COVID-19, which may involve a long recovery time[42]. In contrast, no obvious residual symptoms were left at six months post mild BA.2 BTI, we found that Omicron BA.2 BTIs caused mild impairments to the liver and kidneys, with a recovery time of approximately half a year. Recovery from liver damage has also been shown to be one of the hallmarks of survival in critically ill patients[23]. The results were consistent with a previous report that ~9% of moderate-severe BA.2 infection may lead to symptomatic long COVID at six months post BA.2 infection[13]. Additionally, Age and RNAemia were reported to be the best binary signatures associated with 28-day ICU mortality[23]. However, we did not find

RNAemia in our study. It might be due to infected variant and participant differences, as female, pre-existing medical conditions, and severity during acute infection were risk factors for developing long COVID[13,43,44]. It was reported that SARS-CoV-2 infection can cause lymphopenia[45,46], we also found a low Lym% three months post BA.2 BTI, along with elevated neutrophil and monocyte counts. It is worth noting that NAb production was not influenced. We observed broadly/ durable and potent levels of NAbs against SARS-CoV and SARS-CoV-2 VOCs, which is consistent with previous reports[47–49]. Studies have demonstrated that Omicron BA.1 BTI in BNT162b2, mRNA-1273 vaccines, or inactivated virus vaccines boost serum NAbs titers against VOCs, but not BA.2.12.1 or BA.4/BA.5[50,51]. Here, we found that BA.2 BTIs after 2x Convidecia vaccination induced robust neutralization of not only BA.2.12.1, but also BA.4/5 and BA.2.75, compared with the control group[50]. Sera of BA.2 BTI convalescents showed high levels of neutralization of BF.7 and BQ series regardless of three or six months post BTI, indicating potential protection against these VOCs[52]. Notably, the neutralization efficacies of the XBB series and CH.1.1 were significantly reduced, indicating that immune evasion of XBB, XBB.1.5 and CH.1.1 was significantly enhanced. Therefore, mild Omicron BA.2 BTIs induced potent NAbs against SARS-CoV-2 variants last longer than half a year but cause tissue impairments and require six months for the body to rebuild its defenses against other possible pathogens, as in vaccinated healthy persons.

To further elucidate the above phenomenon, we conducted an in-depth analysis of the recovery of patients six months after BA.2 BTI using proteomics and scRNA-seq techniques. Single-cell transcriptomics and VDJ sequencing have been used to reveal the pathogenesis of COVID-19 in the acute phase and antibody activity in the recovery phase[53–55], however, few studies have applied these to long-term recovery assessment. The long-term effects of COVID-19 on the immune system caused by BTI are largely unknown. With in-depth proteomics and scRNA-seq analyses together with analyzing clinical parameters, we found that the impacts of BA.2 BTI last longer than six months, mechanisms may be illustrated by the following four aspects.

Firstly, the results showed that although there was no significant change in many parameters and the immune repertoire six months post BTI, immune cell levels were still affected in the long term as high levels of cell migration, coagulation, antiviral, interferon, antigen presentation and T cell activation activity were observed. Particularly, *CSRP1, ILK, ITGA2B, F11R, SELP, FCER1G*, and *TFPI* expression levels were upregulated, rescuing coagulation dysfunction. Notably, the proteins coded by *CD47, MFGE8, EPX, F11R, TFPI, ILK, ITGA2B, FCER1G, PHGDH, LMNA* and *FASN* (List of abbreviations can be found in Supplementary Data 7) may play key roles in regulating immunity, coagulation and metabolic pathways. In addition, small PLTs may contribute to the abnormal coagulation profile, which can be further aggravated by liver impairment. Thereafter, the impact on the clotting system lasted longer than six months.

Secondly, six months after BA.2 BTI, the protein synthesis and processing activity of NK cells and monocytes still showed lower levels than the control group. However, high antigen presentation and

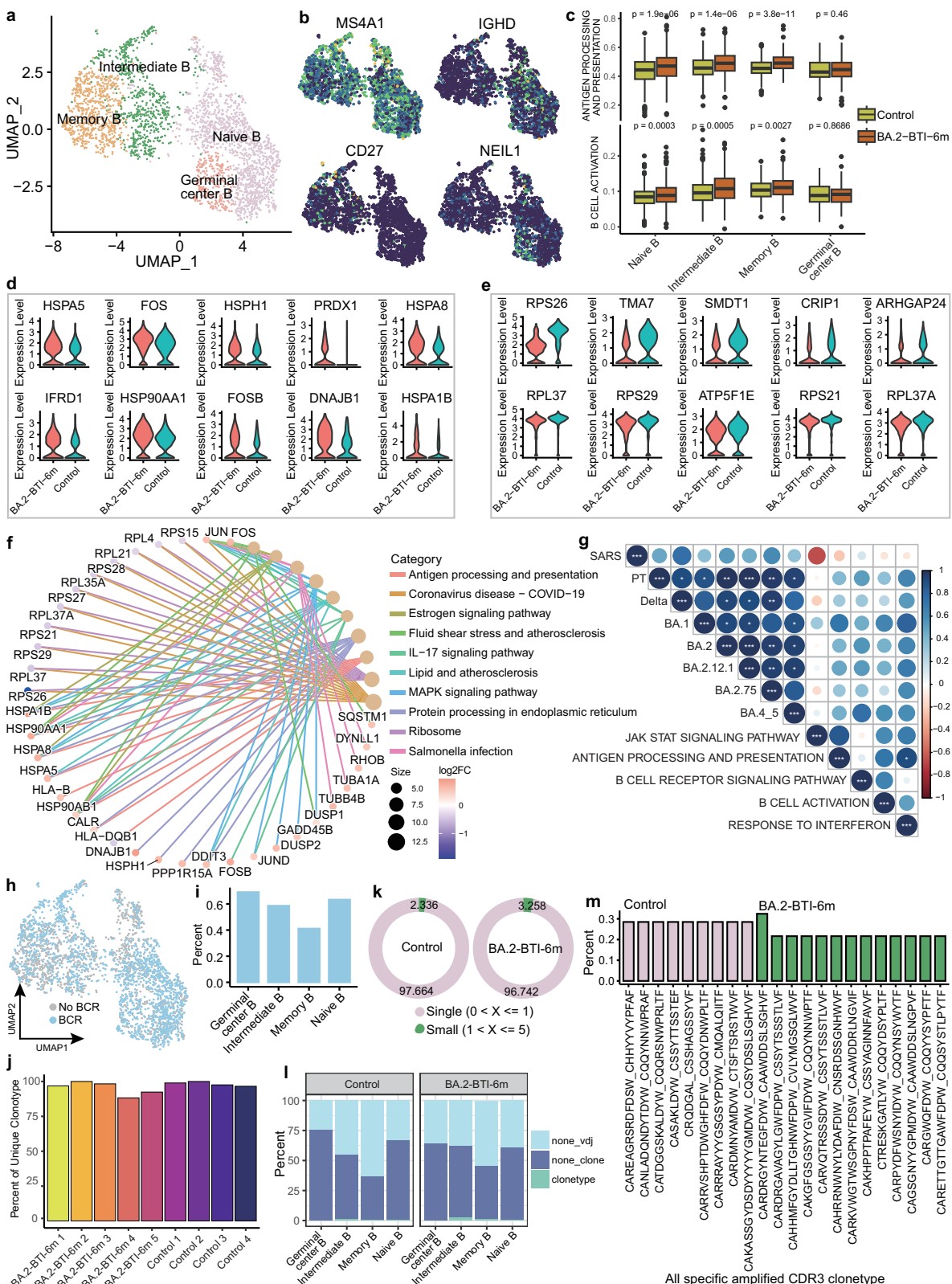

antiviral activity maintained by innate immune cells (except for CD16 Mono) for a long time may ensure that T cells respond quickly to reinfection, especially to maintain the activation and T cell cytotoxicity of memory CD8 T cells. Additionally, although the titers of NAbs against sub-variants XBB, XBB.1.5 and CH.1.1 were limited. The activated T cell and antigen presentation in BTI group may supply some cross-protective effects to these sub-variants because they would react faster, more effectively and not stringently specific. Thus, over the investigating time of six months, the overall defensive immunity against most of the current variants induced by 2x Convidecia vaccinations and BA.2 BTIs did not wane over time.

Thirdly, although the diversity of the T cells, B cells, TCR and BCR returned to normal levels six months after BTI, T and B cells stayed relatively more active in cell activation, differentiation and host

**Fig. 7 | Single-cell sequencing characterized changes in B cell molecules and B cell receptor (BCR). a** UMAP graph displaying subpopulations and clustering of B cells. **b** UMAP map showing expression of marker genes in B cell subpopulation, including naïve B cells (MS4A1$^+$ IGHD$^+$), memory B cells (MS4A1$^+$ CD27$^+$), intermediate B cells (IGHD$^+$ CD27$^+$), and germinal center B cells (MS4A1$^+$ NEIL1$^+$). **c** Box plot showing differences in antigen processing and presentation as well as B cell activation scores of B cell subpopulations between control and BA.2-BTI-6m groups. Minima: Lower limit of the whisker; Maxima: Upper limit of the whisker; Centre: Median line inside the box; The upper and lower box bounds represent the 25% and 75% percentile of data. Two tailed Wilcoxon rank sum test. **d, e** The violin diagram shows the differential genes upregulated and downregulated by top 10 in naïve B cells. **f** Functional enrichment analysis of differential genes by naïve B cells. **g** Correlation between antigen presentation, interferon and B cell activation module scoring and neutralizing activity of different virus strains. ***$p < 0.001$, **$p < 0.01$, *$p < 0.05$. **h** UMAP diagram showing B cells with BCR information. **i** Bar chart showing proportion of BCR detected in B cell subsets. **j** Bar chart indicating BCR clonal proportion of a single sample. **k** The circular plot shows the proportion of BCR with different frequencies in the control and BA.2-BTI-6m groups. **l** Stacked bar chart displaying proportion of clonal subtypes in B cell subpopulations. **m** Bar chart showing most-amplified specific BCR CDR3 sequence in control and BA.2-BTI-6m groups. Pink: control, green: BA.2-BTI-6m.

defence, and the TCR and BCR sequences of specific amplification in the BTI group still existed and the abundance is relatively high in the BA.2-BTI-6m group (Supplementary Fig. 7), especially the TCR with medium- and high-frequency amplification, accounting for a relatively high proportion of the BTI group (Supplementary Fig. 7). In addition, VDJ rearrangements and amplified BCR persisted for six months after BTIs, consistent with the NAb assessments. Interestingly, neutralizing ability was found positively correlated with antigen presentation, interferon, B cell activation and BCR signaling. This means that people in the BA.2 BTI group can fight upcoming infections more strongly than the controls.

Finally, BTIs have a profound and long-term impact on material metabolism, protein synthesis and processing, as overall metabolism and protein synthesis levels declined even six months post BA.2 BTI, undetected by clinical laboratory measures. Therefore, six months after BA.2 BTI, metabolism was downregulated, while the body maintained high antiviral and immune activation activities. Further, from clinical data, we confirmed that levels of prolactin, cortisol and GA remain abnormal even six months post BA.2 BTI. Sex usually contributes to the fluctuations of hormones, like prolactin, especially in females. But in this study, the proportion of males is high (75–80%) and we had matched the sampling date thus sex might contribute little to the differences (Supplementary Tables 1 and 2). Instead, the differences were more likely caused by BA.2 BTI. The mechanism could be an interesting topic to further study as many infected people complained about changes in libido as a symptom of long COVID[9]. It was reported that SARS-CoV-2 could hijack the host mRNA and translation system by adjusting host immunity[56,57]. Hypothetically, prolactin first increased to regulate cellular immunity and humoral immunity[27,28], and went down along with the recovery condition but took a long time. Cortisol, as an immunosuppressive factor[29,30], and homeostatic stress response regulator[31] did not rise during the early stage but increased at six months to regulate the long-term overactive immune status, and moderate-intensity stress caused by COVID-19[31]. Cortisol may also regulate blood glucose level[29,30] to rescue the down-regulated GA. Although this may benefit anti-infection measures, a downregulated metabolism and unrecovered coagulation system may cause other bodily dysfunctions.

In conclusion, our in-depth analysis sheds light on the long-term prognosis of Omicron BA.2 BTI. In particular, coagulation abnormalities alert individuals with aplastic anemia or other blood-clotting disorders to potential infections. Importantly, our results imply a long-term imbalance between anti-pathogen immune activity and metabolism after BA.2 BTI, which may contribute to the pathological mechanism of prolonged COVID-19 development. This also suggests that a balance between immunity and metabolism is required for complete recovery. Additionally, in the absence of obvious symptoms, we do not recommend using any drug to correct the immunity-metabolism imbalance as major manifestations of long COVID caused by mild BA.2 BTI were induced by abnormal mental state[13]. But coagulation disorder should be kept an eye on, and the prolactin level can be interfered with if retained too high, especially for male patients, which can not only reduce inflammation to a certain extent but also relieve related long-term complications.

This study has some limitations. We did not check other possible infections as no recent illness was claimed during sampling. The number of scRNA-seq samples may have been insufficient. A subsequent scale-up or in-depth clinical monitoring is highly encouraged. The study mainly provided a descriptive observation six months after BA.2 BTI, and the mechanism of the imbalance between immunity and metabolism remains to be explored further.

## Methods
### Participants and sampling
This study was approved by the Ethics Committee of the Institute of Microbiology, Chinese Academy of Sciences (SQIMCAS2022127). 2x (Double) Ad5-vectored COVID-19 vaccine (Convidecia, CanSino Bio Co., Ltd.)-immunized individuals who experienced mild Omicron BA.2 BTI - 10 months after the last shot were recruited ($n = 60$, median age = 20 (18–29), 48 males). Among the 60 participants who experienced mild-moderate BTI, four people had lung shadows visible on CT scanning after infection and received antiviral treatments (like Azivudine and Nirmatrelvir Tablets/Ritonavir Tablets(co-packaged)), and non-steroidal anti-inflammatory drugs (NSAIs). The others had no apparent lung abnormalities and were treated symptomatically according to their wishes. At three months after BTI, the abnormal lung CT manifestations of the four people disappeared. Four people (4/60, 6.7%) (not limited to those with abnormal CT) had physical strength, lung function, and vestibular function abnormalities at 3-month post BTI. However, no obvious residual symptoms were left at 6-month post BTI. None of them were immunocompromised before getting COVID-19. Samples were collected around the same time (7–9 a.m.) of a day in a month at three and six months after BTI to minimize the influence of circadian rhythm (BA.2-BTI-3m and BA.2-BTI-6m groups) (Fig. 1, Supplementary Tables 1 and 2). Individuals who had received 2x Convidecia ($n = 20$, median age = 25 (20–41), 15 males) were recruited for this study as controls. The controls were confirmed uninfected by nucleic acid testing. Blood samples were collected approximately 12 months after the administration of the second dose. Informed consent was obtained from all the volunteers. Sex, number and age of participants in clinical routine examinations and pseudovirus neutralizing assay were in line with the above participant information (Supplementary Tables 1 and 2) except for missing values. Participants included in Proteomic ($n = 18$, median age = 23) and single-cell sequencing ($n = 9$, median age = 22) analysis were all male.

### Routine blood examinations
Blood examinations included clinical parameters of the major organs (liver, kidney, heart and blood). For clinical characteristics, we included factors indicating the functions of the liver, kidney, cardiovascular system and hematopoietic system. These routine hospital tests were performed at the Department of Clinical Laboratory of the Air Force Medical Center, China. Liver function evaluating parameters included total bilirubin, direct bilirubin, total serum protein, albumin, albumin/globulin, prealbumin, total bile acid, cholinesterase (CHE), alanine

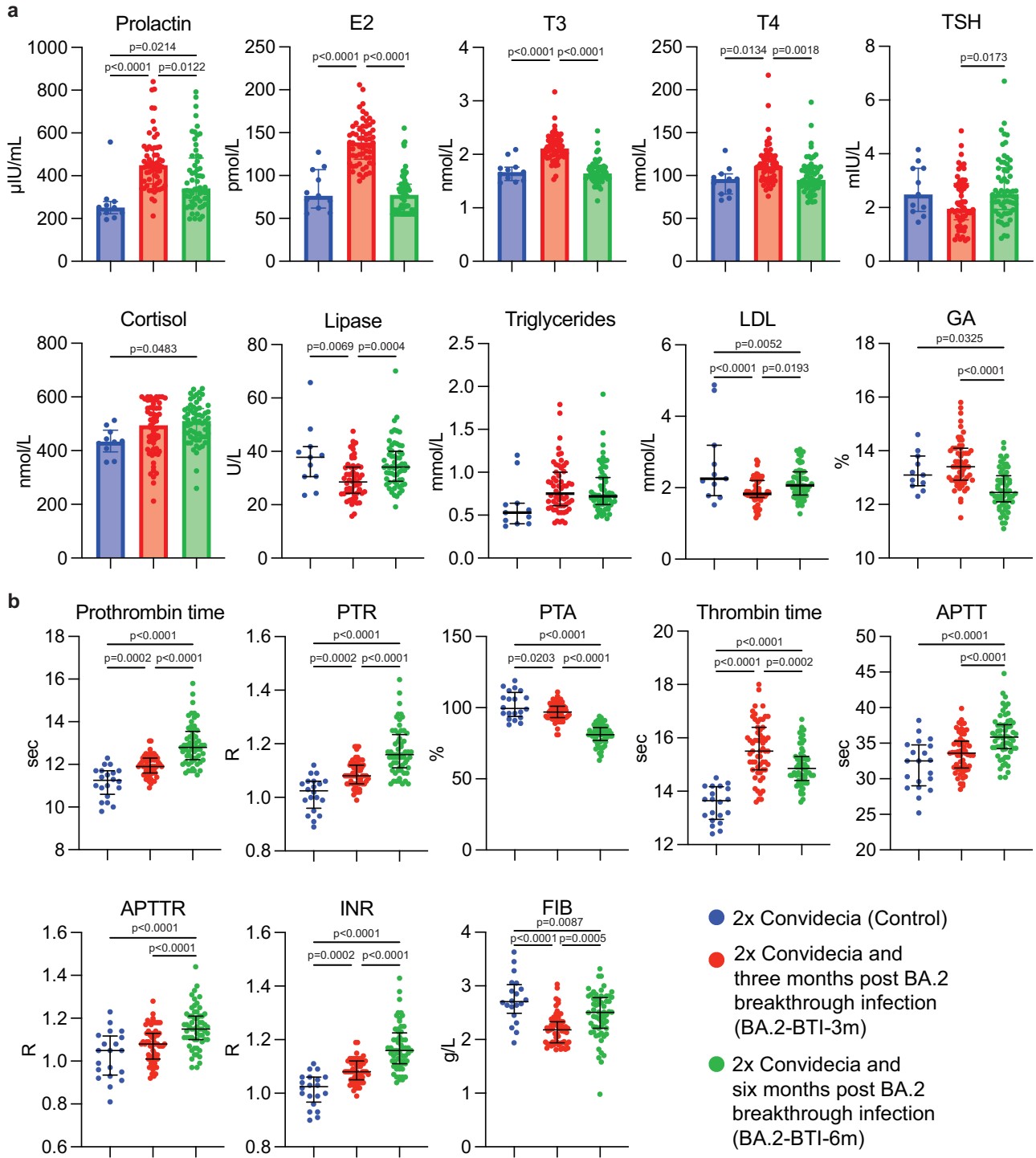

**Fig. 8 | Changes in complementary parameters related to metabolism and immunity, and coagulation profile from BA.2 BTI convalescents. a** Dynamic levels of hormones, lipids and blood glucose (recent two-three weeks) (*n* = 11 biologically independent samples in the Control group, one missing value in Cortisol; *n* = 59 for BA.2-BTI-3m; *n* = 60 for BA.2-BTI-6m). **b** Changes in parameters indicating function of clotting system (*n* = 20 in the Control group; *n* = 59 for BA.2-BTI-3m; *n* = 60 for BA.2-BTI-6m). The Control group is colored in blue. Groups of BA.2-BTI-3m and BA.2-BTI-6m were stratified by time after BA.2 BTI on 2x Convidecia, and colored in red and green separately. E2 estradiol, T3 triphosphothyronine, T4 thyroxine or tetraiodothyronine, TSH thyroxine stimulating hormone, LDL Low-density lipoprotein, GA glycated albumin, PTR prothrombin time ratio, PTA prothrombin time activity, APTT activated partial thromboplastin time, APTTR APTT ratio, INR international standard ratio, FIB fibrinogen. Data are presented as median with interquartile range. *P* values reflect two-sided ordinary ANOVA tests adjusted for multiple comparisons. Source data are provided as a Source Data file.

aminotransferase (ALT), aspartate aminotransferase (AST), γ-glutamyltransferase (GGT), α-L-fucosidase (AFU), hypersensitive C-reactive protein (HS-CRP) and superoxide dismutase (SOD). Electrocyte and renal function evaluation parameters included serum potassium, sodium, chloride, calcium, phosphorus, magnesium (Mg$^{2+}$), blood urea nitrogen (BUN), Creatinine (Cr), serum β2-microglobulin (β2-MG), urobilinogen, uric acid alkalinity (pH), specific gravity and microalbumin in urine. Cardiac function evaluation parameters included homocysteine, myoglobin, creatine kinase (CK), creatine kinase MB isoenzyme (CK-MB) and high-sensitivity cardiac troponin T

(hscTnT) levels. Routine blood tests were also performed to assess the hematopoietic system. Coagulation profile and levels of hormones, serum glucose, lipase and lipid were retrospectively analyzed. Statistical analyses were performed using GraphPad Prism (version 8.4.3; GraphPad Software, LLC). The results are presented as the median with interquartile range (IQR). Statistical significance was determined using the ordinary one-way ANOVA test, and $p < 0.05$ was considered significant.

## Pseudotyped virus neutralization assay

Here, we employed a pseudovirus assay to evaluate NAbs titers in serum samples obtained from patients with BA.2 BTI and those who had been vaccinated twice with Convidecia against pseudoviruses of SARS-CoV, SARS-CoV-2 prototype, Delta and Omicron BA.1, BA.2, BA.2.12.1, BA.2.75, BA.4 and/or BA.5, BF.7, XBB, XBB.1.5, BQ.1, BQ.1.1 and CH.1.1. The pseudotyped virus neutralization assay was identical to that used in our previous work[58,59]. Briefly, pCAGGS was used as a vector to construct S protein expression plasmids for SARS-CoV and SARS-CoV-2 variants (Fig. 2a). Once the pseudoviruses were ready, the serum samples were heat-inactivated at 56 °C for 30 min and then diluted two-fold, starting at appropriate dilutions. An equivalent amount of pseudovirus (1000 transducing units, TU) was then incubated with sera at 37 °C for 1 h. The mixture was then added to pre-plated Vero cells (ATCC CCL81) in 96-well plates. After another 15 h of incubation, the TU numbers were measured using a CQ1 confocal image cytometer (Yokogawa, Japan). Each sample was tested in duplicate, and 50% pseudovirus neutralization titers ($pVNT_{50}$) were derived by non-linear regression using GraphPad Prism (8.4.3). The final results were illustrated using geometric mean titer with a 95% credibility interval. For more details, please refer to the references[58,59].

## Sample preparation for proteome analysis

18 male participants (median age = 23) were included in Proteomic data acquisition and analysis were performed by Shanghai Luming Biological Technology Co., LTD (Shanghai, China) and can be referred to in previous publications[60,61]. Briefly, frozen samples (approximately 100 mg) were rapidly ground in liquid nitrogen to a fine, homogeneous powder and then homogenized in 1 mL of saturated phenol and Tris-HCl (pH 7.5) phenol extraction buffer. After incubation at 4 °C for 30 min, the upper phenolic phase was separated from the aqueous phase by centrifugation, transferred to a new tube, and mixed with five volumes of pre-chilled 0.1 M ammonium acetate-methanol. After overnight incubation at −20 °C, the insoluble proteins were removed by centrifugation, and the pellet was washed twice with pre-cooled methanol and acetone. The pellet was collected and resuspended in lysis buffer. After incubation at -25 °C for 3 h, the supernatant was collected by centrifugation. The total protein in the supernatant was quantified and separated with SDS polyacrylamide gel electrophoresis (12%).

Equal amounts of protein (50 μg) with the same concentration and volume from each sample ware added to dithiothreitol at a final concentration of approximately 5 mM and incubated at 55 °C for 30−60 min. Subsequently, iodoacetamide was added to a final concentration of approximately 10 mM and the mixture was kept at -25 °C in the dark for 15−30 min. Then, six times the volume of pre-cooled acetone was added to the above mixture to precipitate the protein, which was incubated overnight at −20 °C. The precipitate was collected by centrifugation, and 100 μL $NH_4HCO_3$ solution (50 mM) was added to precipitate the protein. A corresponding volume of enzymolysis diluent (protein: enzyme = 50:1 [m/m], 100 μg of protein plus 2 μg of enzyme) was added, and the solutions were incubated for digestion at 37 °C for 12 h or overnight. The enzymatic reaction was stopped by adjusting the pH to 3 by adding phosphoric acid. The samples were desalted on SOLA™ SPE. After drying under vacuum, the samples were resuspended and iRT peptides (1:10) were added. The peptides of the pooled samples were fractionated using a 1100 HPLC system (Agilent, Santa Clara, CA, USA). Mobile phases A (HPLC water with 2% acetonitrile) and B (HPLC water with 98% acetonitrile) were used for reversed-phase gradients. A solvent gradient was set to separate the tryptic peptides at a fluent flow rate of 250 μL/min and monitored at 210 nm. After fractionation, the peptides were lyophilized for mass spectrometry (MS) analysis.

## Data-dependent acquisition (DDA) and DIA mass spectrometry analysis

4D-DIA mass spectrometry (MS) analysis was performed as the samples were separated using a C18 column (25 cm × 75 μm) on an EASY-nLC™ 1200 system (Thermo Fisher, Waltham, MA, USA)[60,61]. The flow rate was 300 nL/min and the linear gradient was set accordingly. The DDA MS data for the library was acquired via the PASEF method as follows: MS data were collected over an m/z range of 100−1700, and during each MS/MS data collection, each TIMS cycle time was 1.1 s; each cycle included 1 MS and 10 MS/MS 100 msec TIMS scans; in each of the 10 PASEF MS/MS scans an average of 12 precursors were selected, resulting in an MS/MS data acquisition rate of 109 Hz. For the DIA, 56 DIA windows were acquired (automatic gain control target 3e6 and auto for injection time), and the collision energy was ramped linearly as a mobility function from 59 eV at 1/K0 = 1.6 Vs cm$^{-2}$ to 20 eV at 1/K0 = 0.6 Vs cm$^{-2}$. The MS/MS spectra were recorded from 100 to 1700 m/z.

## Database search and proteomic analyses

The default factory settings were used for the Spectronaut Pulsar™ 15.3.210906.50606 (Biognosys, Swiss) search and library generation (including trypsyin/P as the enzyme, up to two missed cleavages allowed Oxidation of Me as a variable modification, carbamidomethyl as a fixed modification, and 1% FDR for PSM, peptide, and protein identification). The DDA search results were imported into Spectronaut Pulsar™. The DIA data were analyzed with Spectronaut to search the above constructed spectral library. The main parameters of the software were set as follows: the precursor Q-value cutoff and protein Q-value cutoff were set as 0.01, the Normalization Strategy was set as Local Normalization, and MS2 was used as Quantity MS-Level. The thresholds of fold change (>1.2 or <1.2) and p-value (P) < 0.05 and q-value < 0.25 were used to identify DEPs. All identified proteins were annotated using GO (http://www.blast2go.com/b2ghome; http://geneontology.org/) and Kyoto Encyclopedia of Genes and Genomes (KEGG) pathway analyses (http://www.genome.jp/kegg/). Differentially expressed protein (DEP)s were used further for GO and KEGG enrichment analyses. Protein-protein interaction analysis was performed using the String database (https://string-db.org/).

## Preparation of PBMC single-cell suspension

Nine males (median age = 22) were included in single-cell RNA sequencing, and analysis was performed by OE Biotech Co., Ltd., (Shanghai, China). Peripheral blood mononuclear cell (PBMC)s were stored at −80 °C with serum-free cell freezing solution (Cat. No. C40100) immediately after isolation and transferred to liquid nitrogen after about 24 h for storage. After defrosting, PBMCs were resuspended in 1 mL RPMI 1640 medium (Corning, Cat. No. 10-040-CVR) with 0.04% BSA, and the single-cell suspension concentration and cell viability were determined. If the proportion of dead cells was high, the cells were removed according to the MACS Dead Cell Removal Kit (130-090-101) instructions and measured again. A cell viability >85% allowed single-cell library sequencing.

## Single-cell library sequencing

Nine male samples (including 5 BA.2-BTI-6m and 4 controls) were used for single-cell sequencing. The single-cell suspension was prepared in water for cDNA library amplification using the 10× Genomics

Chromium Next GEM Single Cell 5′ Reagent Kits (version 2.0; Cat. No. 1000165). The Chromium™ Single Cell 5′ Library Construction Kit (Cat. No. 1000020) was used to construct the DNA library. The constructed library was then sequenced using PE150 sequencing on an Illumina Nova 6000 platform. T cell V(D)J and B cell V(D)J enrichment analyses were performed using the 10× Genomics Chromium™ Single Cell V(D)J Enrichment Kit, Human T Cell (Cat. No. 1000005) and Human B Cell (Cat. No. 1000016) according to the manufacturer's instructions. The libraries were amplified using a Chromium TCR amplification kit (Cat. No. 1000252) and BCR amplification kit (Cat. No. 1000253), and the experiment was performed according to the manufacturer's instructions.

## Single-cell expression data generation

The FastQC software was used to evaluate the data obtained to ensure the quality of the raw sequencing data. The raw data were mapped to the human reference genome (GRCh38, https://cf.10xgenomics.com/supp/cell-exp/refdata-gex-GRCh38-2020-A.tar.gz) using CellRanger, which is a 10x genomics software that labels different mRNA molecules within each cell by identifying the barcode and UMI sequences for single-cell transcriptomic quantification.

## Single-cell immune repertoire data generation

The UMI screening standard was supported by 400 paired reads. The reads that passed the mapping and UMI standards were used for contig splicing. Validity screening was also conducted on the barcodes and annotations were provided to remove the error information caused by artificial products. The concatenated contig was annotated and screened further to obtain a consensus sequence supported by the sample CDR3. Clonotype typing was performed based on CDR3 amino acid sequences of the obtained samples with consistent sequences.

## Single-cell data analysis

Single-cell data were integrated and clustered using the Seurat R package (version 4) (https://satijalab.org/seurat/). A total of 124,541 cells were obtained from single-cell sequencing of the nine samples, and 108,306 cells remained after quality control. The cell quality control was conducted as follows: cells with a mitochondrial gene ratio exceeding 10% were removed, and only cells with gene numbers ranging from 500 to 4500 and UMI numbers ranging from 800 to 16,000 were retained. DoubletFinder R package (https://github.com/chris-mcginnis-ucsf/DoubletFinder) was used to remove potential doublets, and further manually remove potentially marginalized doublets based on known classic markers. The filtered data were then standardized and normalized, and principal component analysis was performed on the top 2,000 genes with the highest coefficients of variation. The Harmony R package (https://github.com/immunogenomics/harmony) and the anchor module of Seurat were used to remove inter-batch effects between the samples and groups for cell clustering. Based on the elbow point and significance of the different principal components, the top 30 PCs were selected for subsequent cell clustering, and different resolutions were set to determine the cell clusters. Dimensionality reduction and visualization of single cells were performed using the Uniform Manifold Approximation and Projection (UMAP).

## Cell type annotation

Using UMAP, all cells underwent dimensional reduction and were clustered in a two-dimensional space based on shared features. Firstly, the Azimuth algorithm was used to map the data to the reference cell set of PBMC, and then combined with specific high expression genes to manually determine the cell type. Specifically, classic biomarkers for specific cell types were used to identify the cells in different clusters. The *FindAllMarkers* function in Seurat was used to identify the 50 most highly expressed genes in each cluster of cells, providing a comprehensive understanding of cell types based on the top gene and literature. When clustering for the first or the second time, clusters expressing two or more classic markers and marginalized cells were considered doublets and excluded from subsequent analysis.

## Cell difference abundance analysis

Use Milo algorithm[62] to divide the cells of the control group and BA.2-BTI-6m group into different neighborhoods and calculate their spatial distribution differences, mapping them to different cell types. The key parameters for executing the Milo algorithm are k = 10 and d = 30. In addition, the proportion of cell types for each sample was calculated based on the conventional cell percentage and their differences between groups were calculated using the rank sum test.

## Differential gene identification and functional analysis

The *Findmarkers()* function in the Seurat package was used to identify differentially expressed gene (DEG)s between distinct cell groups, using a standard of |logFC|> 0.25 and FDR < 0.01. DEGs only contain genes expressed in at least 25% of cells of the control group or infection group. The ClusterProfiler R package facilitated Gene Ontology and KEGG enrichment analyses and visualization of DEGs.

## Gene set activity score of individual cells

The *AddModuleScore()* function of Seurat was used to calculate the activity scores of different gene sets in single cells. The gene set was sourced from the msigdb R package (Antigen processing and presentation (hsa04520), JAK_STAT_signaling (hsa04630), B cell activation (GO:0042113), B cell receptor signaling (GO:0050853), positive regulation of Treg activity (GO:0045591), response interferon (GO:0034341), protein processing (GO:0016485) and coagulation regulation (GO:0007597, GO:0050819, GO:0050820, GO:0050818). T cell toxicity activity was defined by the following gene sets: PRF1, IFNG, GNLY, NKG7, GZMB, GZMA, GZMH, KLRK1, KLRB1, KLRD1, CTSW, and CST7. The tissue specific gene set based on proteomics comes from the research of Gutmann et al.[23] and Li et al.[24].

## TCR/BCR analysis

Using human GRCh38 as the reference genome, the Cell Ranger vdj pipeline was used to identify the TCR/BCR clonotype and quantify VDJ gene expression. For TCR, we only retained cells with at least one productive TCRα chain (TRA) or TCRβ chain (TRB) for subsequent analysis. Where a cell had two or more paired TRA or TRB chains, we only retained the one with the highest basal expression. Clonotypes were defined based on their unique CDR3 amino acid sequence, and each unique TRA/TRB/TRA-TRB pair was defined as a clonotype. For BCR analysis, we retained only cells with at least one productive heavy chain (IGH) and IGK/IGL for subsequent analysis. When a cell had two or more paired IGH or IGK/IGL chains, only those with the highest basal expression were retained. Each unique pair IGH-IGK/IGL was defined as a clonotype. The scRepertoire R package (https://github.com/ncborcherding/scRepertoire) was used to analyze the single-cell immune repertoire and calculate the clonal diversity of the samples based on the aroma index. Based on the cell barcode information, clonotypes with TCR or BCR were mapped onto the cell UMAP map.

## Reporting summary

Further information on research design is available in the Nature Portfolio Reporting Summary linked to this article.

# Data availability

The single-cell sequencing data generated in this study have been deposited in the Genome Sequence Archive[63] database under accession code HRA004484 (https://ngdc.cncb.ac.cn/gsa-human). The raw single-cell sequencing data are protected and restrictedly available due to data privacy laws. The processed single-cell sequencing data are

available at the Gene Expression Omnibus database (access number: GSE240694). The mass spectrometry proteomics data generated in this study have been deposited to the ProteomeXchange Consortium via the iProX partner repository under the accession code PXD044441 (http://proteomecentral.proteomexchange.org). The manuscript did not generate original code and the analysis process link can be found in the manuscript methods or contacted by the authors. Source data are provided with this paper.

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

## Acknowledgements

This study was supported by the National Key R&D Program of China (2020YFA0907102, G.F.G.), the National Natural Science Foundation of China (82170004, Y.M. and 82222040, X.Z.), the Youth Innovation Promotion Association of the Chinese Academy of Sciences (2020092, X.Z.), CAS Project for Young Scientists in Basic Research (YSBR-083, X.Z.), the China Postdoctoral Science Foundation (No. 2023M743729, Yanhua L. and 2022M723344, S.Qin) and the Air Force Special Medical Center Science and Technology Booster Program (2022ZTYB39, L.D.). The authors thank all the participants recruited for their contributions to this study. We sincerely thank Dr. Daniel Montiel-Garcia (Ph.D.) from Scripps Research and Mohammed H. Althunayan (MPH) from Michigan University for their time and kind help in language editing and smoothing the manuscript. The authors sincerely thank OE Biotech Co., Ltd., (Shanghai, China) for providing single-cell RNA-seq and TCR/BCR sequencing analysis.

## Author contributions

G.F.G., Y.M., X.Z., Yanhua L. and S.Qin conceptualized the study. Yanhua L. and S.Qin wrote the original draft. G.F.G., Y.M. and X.Z. supervised the study. Yanhua L. and S.Qin did data analysis and figure making with the assistance of S.Qiao. L.D., S.Quan, Ying L., F.F. and Yanhua L. were responsible for sample collection and clinical data assembly and assessments on pathological features. Yanhua L. and D.Y. interpreted the pathological characteristics. Yanhua L., S.Qiao, X.W., P.G. and Y.H. did pseudovirus neutralization assays. G.F.G., Y.M., X.Z., Yanhua L., S.Qin and S.Qiao curated the data. G.F.G., Y.M., X.Z., Yanhua L. and L.D. are responsible for funding resources. All authors reviewed and edited the manuscript. All authors had full access to all the data in the study and had final responsibility for the decision to submit for publication.

## Competing interests

The authors declare no competing interests.
