## [Peer Review File · Nature Communications]

Long-term effects of Omicron BA.2 breakthrough infection on immunity-metabolism balance: a 6-month prospective studyReviewers' Comments:

Reviewer #1:

Remarks to the Author:

In this study, Li and colleagues conducted a cohort study evaluating immune correlates for Omicron BA.2 breakthrough infection compared to vaccinated and uninfected participants. It is a nice prospective cohort study, and the authors leveraged multiple modalities to thoroughly evaluate immune correlates to BA.2 breakthrough infection and demonstrated certain differences in proteomics and PBMC transcriptomics. My comments are below:

Major

General

- The authors mentioned "long Covid" in the abstract and discussion. However, it is unclear to me whether those participants with breakthrough infection all experienced "long Covid" and how symptomatic they were (would use certain symptom score or diary), and how long did the residual post-acute symptoms last. To me, those participants were just experiencing breakthrough BA.2 infection without long-lasting residual symptoms. If those participants truly had "long Covid" or PASC, I would at least document the symptoms they had and duration based on this recently published framework (PMID 37278994).
- For those control participants, understandably that the authors did the nucleic acid test to confirm that they did not have *active* infection at time of blood draw. How would the authors rule out the possibility of asymptomatic or paucisymptomatic infection within 3 months prior to the blood draw, which may affect the immune features you were measuring? I would at least check anti-N antibodies and if you want to be stringent about it, I would only include those who had negative anti-N as control, or at least do sensitivity analyses restricting to those with negative anti-Nucleocapsid ab.

Results

- Participant characteristics were listed in Table S1. Did any of the participants receive antiviral treatment, convalescent plasma, steroid, anti-IL6, or monoclonal antibody products? Were any of the participants immunocompromised before COVID. This would be very important to know.
- For the neutralization data, it is impressive that the authors tested neutralization against a great variety of variants/subvariants.
- Figure 3: For the proteomic data, did you see any up or down-regulations of any cytokines or chemokines? In addition, can the authors perform tissue gene-set enrichment analyses and see if there are specific tissue damage signals (e.g. lung-specific protein sets, liver-specific etc.). Previous work has shown that in acute COVID, severe disease and RNAemia are associated with certain lung tissue damage and coagulation disturbance signal (PMID: 34099652, 34196300 etc).
- Also for the proteomic data, were you able to detect any Spike or other SARS-CoV-2 protein, as persistent antigenemia has been reported to be related to post-acute symptoms (<https://doi.org/10.1093/cid/ciac722>).
- For Figure 5, did the authors specifically look at NK effectors expression (e.g. granzymes, perforin).
- For Figure 5, I am curious if the authors could apply previously defined framework for monocyte function/phenotype (e.g. MS1 phenotype PMID: 34103408) to current analysis and see if there is any differences in MS1 module etc.
- For T cell signature, were the authors able to identify any differences in Treg activity or function (<https://www.pnas.org/doi/10.1073/pnas.2111315118>). Merely from cytotoxic function, both groups seemed comparable and control group might have even stronger cytotoxic activity; however Treg is not quite known for this and I'd like the authors to show if there are any suppressive activity that is different between those two groups.
- For clonal T and B cells, is clonality associated with differential gene expression? Especially at the same cell subtype level.

- Figure 8: There has been a huge body of evidence on hypercoagulability and COVID. However, in this study, it puzzled me that the thrombin time, PT/INR, and APTT were prolonged after the infection. Were the authors able to measure D-dimer and fibrinogen degradation products and see if there was any evidence of microangiopathy? Also did the authors have data on autoantibody, especially lupus anticoagulant?

- Were you able to confirm that some of the upregulated DEPs from proteomics results were also upregulated in scRNA-seq dataset? I am particularly interested in platelet-related proteins (e.g. ITGA2B, selectin-P/SELP).

Methods

- What time of the day was the blood sample obtained for cortisol? The authors mentioned "same time of a day" but did not specify when. Ideally you want all participants to get bloodwork in the morning, as 8-9am cortisol levels would be more meaningful.

Minor

- Line 107-108 "This suggests the active clearance of exogenous antigens from the body during the first three months after BA.2 BTI" not sure if this is true regarding elevated monocyte %.

- I am just curious if you see any correlation between B cell activation/IFN related signal/antigen processing from scRNA-seq (e.g. antigen processing module score, IRF7, ATF3 etc.) and neutralization levels to different subvariant. Would be interesting to make a correlation heatmap or something equivalent to link pseudovirus neutralization levels and B cell markers at individual participant levels.

- For the discussion of cortisol, it is also associated with stress levels (for example <https://academic.oup.com/jcem/article/90/8/4579/3058896>).

Reviewer #2:

Remarks to the Author:

Li et al. investigate proteomic and transcriptional landscape of blood samples after breakthrough infections (BTI) with the Omicron variant of COVID-19 with up to 6 month follow up. They measured clinical blood parameters for liver, renal and hematopoietic systems for 60 patients with BTI and 20 controls. They also perform virus neutralisation assays, blood proteomics and single cell transcriptomics on a subset of patients. The study is largely descriptive, with limited donors profiled in depth for example only 9 donors profiled with single cell transcriptomics. Although the authors discuss long COVID in text, a key piece of data missing is the symptomatic landscape of these patients with omicron BA.2 BTI. In addition, there have been many studies of long COVID/post-acute sequelae of COVID (PACS), albeit without the context of BTI infection, which the authors do not contextualise/contrast their findings to. Expanding on this, I have some specific comments below:

1. The serology data is the most comprehensive in terms of patient numbers, are the authors able to quantify symptoms of long COVID in these patients (eg. from patient data or surveys) to further stratify those with BTI and long COVID symptoms vs those without?

2. Did the authors check for other infections/recent illness history?

3. Figure 2 shows increased neutralisation after break through infection – this is known in other vaccination settings albeit to an older set of variants, this study should be cited:

<https://www.science.org/doi/10.1126/scitranslmed.abn8057>

4. When describing figure 3d and c, the authors refer to "core hub" proteins but it is unclear what this means.

5. How exactly did the authors identify that there were no batch effects in their single cell data integration? Did they test or bench mark different batch correction methods?

6. For single cell data overview, could the authors distinguish CD4 and CD8 T cells? I understand T cells are further subclustered in Figure 6, but this would be useful in the global analysis presented in Figure 4.
7. The authors assess differences in cell percentages, there are more sophisticated methods that could be used instead for example those summarised in:
<https://www.biorxiv.org/content/10.1101/2023.02.24.529894v1.full> using a method such as Milo is independent of cell labels and may reveal hidden differences.
8. For differential gene expression analysis, a more robust method is to perform pseudobulking per cell type/donor and differential expression using bulk tools such as edgeR.
9. For the T cell annotations in Figure 6 the marker genes are not so distinct, could the authors display additional marker genes especially for NKT and CD8 subsets with one defining marker gene. It could also be good to compare the authors annotations to existing annotation from PBMC single cell data, for example using tools like Azimuth, CellTypist or integrating with existing PBMC/covid atlases.
10. How did they calculate scores for antigen presentation, protein processing and cytotoxicity in the single cell data?
11. Without increasing the number of patients in the single cell data, perhaps the authors could integrate their data with other single cell sequencing data from COVID patients followed longitudinally? For example <https://insight.jci.org/articles/view/165299>
12. For the TCR and BCR repertoire – do they have this for 3 months post BTI? If so, could they incorporate it into the analysis.
13. Again for the TCR and BCR data – did the authors identify any shared clones between individuals? For example, were there more shared clones in BTI individuals than controls?

Minor comments:

1. In the abstract the abbreviation PLT is used, please define this in the abstract.
2. Please add the numbers of patients for each assay/data type in text and figure legends, this information was difficult to find.
3. In the text relating to Figure 6, the authors state that “Memory T cells levels undergo intense and lasting dynamic shift”. I believe the authors here are referring to differentially expressed genes rather than T cell abundance (as the term level suggests). Regardless, I would say that this is an overstatement of the data.

Reviewer #3:

Remarks to the Author:

In the manuscript by Li et al. entitled “Long-term effects of Omicron BA.2 breakthrough infection on immunity-metabolism balance: a 6-month prospective study” the authors recruited individuals who received a vaccine against COVID-19 (2x Double Ad5-vectored COVID-19 vaccine; Convidecia, CanSino Bio, 398 Co., Ltd.) and developed a mild breakthrough infection (BTI) ~10 months after the last shot (n=60), and compared them to individuals who received the same vaccine but did not develop a BTI (n=20). Blood from individuals from the BTI group was collected 3 months and 6 months post infection; which corresponds to ~13 months and 16 months after the second vaccine shot, respectively. Blood from the control group was collected ~12 months after the second vaccine shot. The authors applied mass spectrometry-based proteomics of serum, scRNA-seq of PBMCs, and analysed clinical variables for a comparative analysis. The authors found differences including in coagulation, metabolism and anti-pathogen immunity.

Although the manuscript is methodically solid and well-written, the study allows limited conclusions because: i) the study is limited to relatively few patients and some experiments rely on few samples (e.g. proteomics of 18 samples; scRNA-seq of 9 samples); ii) the findings rely heavily on

bioinformatical analyses with little biological validation; iii) no outcome data or information on development of long COVID is contained. The manuscript may therefore be better suited for lower-tiered journals of the Nature family such as Scientific Reports.

Comments to the authors

- Please discuss the following limitations: i) the study is limited to relatively few patients and some experiments rely on few samples (e.g. proteomics of 18 samples; scRNA-seq of 9 samples); ii) the findings rely heavily on bioinformatical analyses with little biological validation; iii) no outcome data or information on development of long COVID is contained.
- Please provide a table that compares the baseline characteristics of the control and BTI groups.
- Protein Abbreviations should be explained either in the text or in a table.
- The volcano plot in Figure 3b appears to only show nominal p-value. Please also report results adjusted for multiple testing.
- Please explain the rationale for looking at PBMCs.
- The finding of liver impairment is interesting and has already been found in the acute setting based on proteomics data (DOI:10.1038/s41467-021-23494-1) as well as circulating microRNA data (miR-192: DOI:10.1016/j.trsl.2021.05.004 ; miR-122 DOI:10.1093/cvr/cvab338) and should be discussed.
- The authors should also discuss the role of RNAemia in the context of long COVID.

Reviewer #1 (Remarks to the Author):

In this study, Li and colleagues conducted a cohort study evaluating immune correlates for Omicron BA.2 breakthrough infection compared to vaccinated and uninfected participants. It is a nice prospective cohort study, and the authors leveraged multiple modalities to thoroughly evaluate immune correlates to BA.2 breakthrough infection and demonstrated certain differences in proteomics and PBMC transcriptomics. My comments are below:

Major

General

1. The authors mentioned long Covid in the abstract and discussion. However, it is unclear to me whether those participants with breakthrough infection all experienced long Covid and how symptomatic they were (would use certain symptom score or diary), and how long did the residual post-acute symptoms last. To me, those participants were just experiencing breakthrough BA.2 infection without long-lasting residual symptoms. If those participants truly had long Covid or PASC, I would at least document the symptoms they had and duration based on this recently published framework (PMID 37278994).

Response: Thank you for the advice. We have added symptomatic information of the participants in the manuscript (Materials and methods section). We followed-up symptom information of the participants with phone call and clinical materials, collecting if they have long-term symptoms. Among the 60 participants who experienced mild-moderate BTI, four people had lung shadows visible on CT scanning after infection and were treated with antiviral + non-steroidal anti-inflammatory drugs. The others had no apparent lung abnormalities and were treated symptomatically according to their personal wishes. At three months after BTI, the abnormal lung CT manifestations of the four people disappeared. Only four people (4/60, 6.7%) (not limited to those with abnormal CT) had physical strength, lung function, and vestibular function abnormalities at three months after BTI, though, no obvious residual symptoms were left at 6 months post BTI. Additionally, the definition of long COVID specified the long-term effects from SARS-CoV-2 infection and signs, symptoms and conditions, indicating long COVID is not only focused on symptoms but also the long-term effects of SARS-CoV-2 infection. Meanwhile, we understand the common understanding of symptomatic long COVID among the public. Thus, to avoid misunderstanding we have made some adjustments in the manuscript as well.

2. For those control participants, understandably that the authors did the nucleic acid test to confirm that they did not have *active* infection at time of blood draw. How would the authors rule out the possibility of asymptomatic or paucisymptomatic infection within 3 months prior to the blood draw, which may affect the immune features you were measuring? I would at least check anti-N antibodies and if you want to be stringent about it, I would only include those who had negative anti-N as control, or at least do sensitivity analyses restricting to those with negative anti-Nucleocapsid ab.

Response: Thank you for the valuable suggestion. We understand the concern. The control samples were collected when China was using the “Zero Covid” policy. During those days, Chinese people were under intensive nucleic acid testing, sometimes daily and sometimes twice

in three days. More importantly, under this policy, all people who have ever been infected were carefully recorded. In this case, we could distinguish the infected and the uninfected people regardless of very mild illness or asymptomatic infection. That is also why we were certain the controls never experienced SARS-CoV-2 infection upon blood draw.

Results

1. Participant characteristics were listed in Table S1. Did any of the participants receive antiviral treatment, convalescent plasma, steroid, anti-IL6, or monoclonal antibody products? Were any of the participants immunocompromised before COVID. This would be very important to know. Response: Thank you for the valuable suggestion. We have updated this information in the text. Four of them, with abnormal lung imaging of CT scanning after BTI, received antiviral treatments (like Azivudine and Nirmatrelvir Tablets/Ritonavir Tablets(co-packaged)) and NSAIDs. Most of the participants received symptom-specific treatments, like Lianhua Qingwen capsules, ibuprofen, as their wishes. None of them were immunocompromised before COVID.

2. For the neutralization data, it is impressive that the authors tested neutralization against a great variety of variants/subvariants.

Response: Thank you for the comment. We tried our best to cover the major circulating variants of concern.

3. Figure 3: For the proteomic data, did you see any up or down-regulations of any cytokines or chemokines? In addition, can the authors perform tissue gene-set enrichment analyses and see if there are specific tissue damage signals (e.g. lung-specific protein sets, liver-specific etc.). Previous work has shown that in acute COVID, severe disease and RNAemia are associated with certain lung tissue damage and coagulation disturbance signal (PMID: 34099652, 34196300 etc).

Response: Thank you for the comment. Our proteomics detected cytokines/chemokines such as IL36G, IL1RN, TNFSF13, TNFAIP8, TNFAIP6, TNFAIP8L2, MIF, CXCL3, CXCL12, CCL5, CCL24, CCL21, CCL18, CCL16, CCL14, etc. Most of them are not differentially expressed, except for IL1RN (Interleukin-1 receptor antagonist protein) which was significantly downregulated in the BA.2-BTI-6m group and MIF (Macrophage migration inhibition factor) which was significantly upregulated.

In addition, based on your suggestion, we conducted GSEA analysis of the liver, lung, pancreas, heart specific protein gene sets, and coagulation system to reflect potential tissue damage and hematological changes. The results showed that coagulation in the BA.2-BTI-6m group showed an activated state ($p=0.0012$). The GSEA enrichment results of tissue specific markers were not significant, indicating no prominent tissue damage to the liver, lungs, pancreas, or heart, which is consistent with clinical result. Thank you.

Figure R1. GSEA analysis of tissue proteome specific genes. A: The GSEA map shows the enrichment of coagulation related genes and proteome specific genes in liver, lung, heart, and pancreatic tissues. B: The intersection and difference between coagulation related genes, liver, lung, heart, and pancreatic tissue specific genes, and measured proteins. Each column

represents the intersection of tissue specific protein gene sets and measured proteins, while * represents a significant difference between the control group and BA.2-BTI-6m.

4. Also for the proteomic data, were you able to detect any Spike or other SARS-CoV-2 protein, as persistent antigenemia has been reported to be related to post-acute symptoms (<https://doi.org/10.1093/cid/ciac722>).

Response: Thank you for the comment. According to your suggestion, we mapped the proteomics data to the protein sequence library of SARS-CoV-2. The results showed no protein sequence was mapped to SARS-CoV-2 proteins, meaning that these participants did not develop viremia or antigenemia, which was consistent with clinical results, as no participants showed symptoms 6 months after infection.

5. For Figure 5, did the authors specifically look at NK effectors expression (e.g. granzymes, perforin).

Response: Thank you for the comment. We have supplemented the analysis of the expression changes of NK cell effector proteins such as granzyme, perforin, etc. The results showed that GZMH and GNLY were significantly upregulated in the BA.2-BTI-6m group, while other GZMB, PRF1 and IFNG were significantly downregulated. It should be pointed out that although significant, all logFCs were between 0.25 and 0.5, indicating that after 6 months of recovery from BA.2 BTI, the killing effect of NK cells has become mild. We have also updated the manuscript.

Figure R2. The expression changes of effector proteins in NK cells in BA.2-BTI-6m and

control groups. A: The expression of effector proteins in NK cells in BA.2-BTI-6m and control groups. B: The expression of effector proteins in NK cells in various samples.

6. For Figure 5, I am curious if the authors could apply previously defined framework for monocyte function/phenotype (e.g. MS1 phenotype PMID: 34103408) to current analysis and see if there is any differences in MS1 module etc.

Response: Thank you for your comment. Based on your suggestion, we have re-clustered CD14 monocytes into 4 clusters of cells. By calculating the specific expression genes of each cluster, we found that cells similar to the MS1 are in cluster 0, with high expression of MS1 markers such as ALOX5AP, RETN, THBS1, etc. In addition, cluster 0 also expresses genes such as S100A8 and S100A9, which are highly expressed genes in MS1 compared to MS2 and MS4. We also calculated the scores of the top 50 highly expressed genes in MS1 (DOI: 10.1038/s41591-020-0752-4) and found that they also showed significant enrichment in cluster 0. In summary, we believe that cluster 0 stands for potential MS1 cells. We identified 202 differentially expressed genes of MS1 in control and BA.2-BTI-6m groups. The top upregulated genes in the BA.2-BTI-6m group included TMEM176B, TMEM176A, HLA-DQA1, CD52, DNAAF1, etc., while the downregulated genes were TMA7, AC020656.1, SMDT1, DDIT4, etc. Further functional enrichment analysis showed that the MS1 cells in the BA.2-BTI-6m group exhibited active antigen presentation, T cell activation, and regulation of hematopoiesis, while cytoplasmic translation, ribosome biogenesis and ATP metabolic process were in an inhibitory state. For more details, please refer to Revised Figure 5. Thank you.

Figure R3. Identification of MS1 related cell populations. A: Re-cluster CD14 monocytes into 4 clusters, and the 0th cluster may have an MS1 related phenotype. B: The UMAP map displays the expression levels of markers RETN, ALOX5AP, and THBS1 in MS1 cells, as well as the gene set scoring of the top 50 specific genes of MS1 identified in previous studies. C:

The volcano map shows the DEGs of MS1 cells in the BA.2-BTI-6m and Control groups, totaling 202. D: Functional enrichment analysis of DEG in MS1.

7. For T cell signature, were the authors able to identify any differences in Treg activity or function (https://www.pnas.org/doi/10.1073/pnas.2111315118). Merely from cytotoxic function, both groups seemed comparable and control group might have even stronger cytotoxic activity; however Treg is not quite known for this and I'd like the authors to show if there are any suppressive activity that is different between those two groups.

Response: Thank you for your comment. We have added analysis of Treg cells. As shown in the following Figure A, we did not find any differences in the marker gene of Treg between the control and BA.2-BTI-6m groups. A global differential analysis identified 51 DEGs (Figure B). Further functional analysis showed that these DEGs were mainly involved in biological processes such as RNA splicing, response to unfolded proteins, regulation of protein ubiquitination, and phosphatase activity (Figure C). We also visualized the expression level of Treg activity inhibitory genes and found that IRF1 was significantly upregulated in the BA.2-BTI-6m group (Figure D). In addition, there was no significant difference in the genes associated with increased Treg activity and there was no significant difference in gene set scores (Figure E). For more updated results related to T cells, please refer to the revised manuscript. Thank you.

Figure R4. Analysis on Treg cells.

8. For clonal T and B cells, is clonality associated with differential gene expression? Especially at the same cell subtype level.

Response: Thank you for your comment. We have supplemented differential expression analysis of T cells related to amplification. Considering that the expanded T cells mainly concentrate on CD8 effector memory T cells (CD8 TEM). Therefore, we use CD8 TEM as an example to study the differences between amplified and non-amplified T cells after 6 months of infection. As shown in the figure below, we identified a total of 132 DEGs. Compared with amplified T cells, the significantly upregulated top 10 genes in amplified T cells were mainly GZMB, GNLY, LGALS1, FGFBP2, KLRD1, PRF1, S100A6, S100A4, GZMH, ADGRG1, while the downregulated top 10 genes were GZMK, LTB, DNAJB1, CMC1, IL7R, TCF7, DUSP2, HSPA1B, COTL1, RGCC. It can be seen that amplified T cells upregulated T cell killing effector proteins such as granzyme (GZMB, GZMH), perforin (PRF1), and granulysin (GNLY), and downregulated naive T cell markers such as LTB, IL7R, TCF7, etc. Further functional enrichment analysis showed that upregulated genes were mainly involved in T cell activation, T cell differentiation, and host defense, while downregulated genes were not only enriched in T cells, but also involved in hematopoietic regulation, calcium ion response, and response to topologically incorrect proteins and unfolded proteins. These results suggest that there is a difference in the function of amplified and non-amplified T cells after 6 months of BA.2 BTI.

In addition, due to the small number of B cells and the lower proportion of expansion, we will no longer perform differential analysis related to expansion of B cells.

Figure R5. Differences between amplified and non-amplified T cells. A: Top 10DEGs of amplified and non-amplified T cells. A: Functional enrichment analysis of DEGs in expanded and non-amplified T cells.

9. Figure 8: There has been a huge body of evidence on hypercoagulability and COVID. However, in this study, it puzzled me that the thrombin time, PT/INR, and APTT were

prolonged after the infection. Were the authors able to measure D-dimer and fibrinogen degradation products and see if there was any evidence of microangiopathy? Also did the authors have data on autoantibody, especially lupus anticoagulant?

Figure R6. Serum levels of D-dimer and fibrinogen degradation products

Response: Thank you for the valuable suggestion. We have updated the manuscript. It is true that hypercoagulability and COVID were proved correlated but the samples in this study were collected three months and six months after infection. It is quite understandable that coagulation substances are consumed in large quantities during the infection period (hypercoagulability state) and

enter a state of deficiency of coagulation substances. The principle is similar to the occurrence of disseminated intravascular coagulation (DIC)(doi: 10.1056/NEJM199908193410807). Because it is a self-limiting disease caused by viral infection, the coagulation function would likely recover as the body recovers, but it may take longer. Yes, we have measured D-dimer and fibrinogen degradation products(only the data of three months post BTI were collected), and no significant differences were found (Figure R6). More importantly, fibrinogen (FIB) concentration was reported represent a more useful marker of hypercoagulability than D-dimer (DOI: 10.1177/0885066621997039) that was why we did not present the result of D-dimer in our study. We have also tested autoantibodies when collecting BA.2-BTI-3m samples, including Antinuclear antibody, Antiribonucleoprotein antibody, Anti-Sm, Anti-SSA, Anti-Ro-52, Anti-SSB, Anti-Scl-70, Anti-PM-Scl, Anti-Jo-1, Anti-centromere protein B antibody, Anti-proliferating cell nuclear antigen antibody, Anti-double-stranded DNA, etc., very few positive case were found (Table R1).

10. Were you able to confirm that some of the upregulated DEPs from proteomics results were also upregulated in scRNA-seq dataset? I am particularly interested in platelet-related proteins (e.g. ITGA2B, selectin-P/SELP).

Response: Thank you for your comment. We examined the intersection of DEPs and DEGs and found a total of 29 genes. Among them, 13 genes were found consistently downregulated including RPS15A, S100A11, RPS3, ATP5MG, RPS16, COX4I1, HSP90B1, NAMPT, RPL13A, RPL23, S100A9, CSTA, HNRNPA3. While the genes with consistent upregulation have not been identified. In addition, there were 16 genes with inconsistent protein and mRNA changes. This may be related to the translation regulation and modification of mRNA.

For ITGA2B and SELP (related to platelet activation and coagulation), we only found that they were upregulated in the proteomic level rather than in single cells, which may be due to the scarcity of platelets, which were filtered during the separation of PBMC.

Table R2. Intersection of DEPs and DEGs.

gene	p_val	avg_log2FC	pct.1	pct.2	p_val_adj	cell_type
HSP90AB1	5.64E-16	0.30842056	0.884	0.866	2.07E-11	B
UBB	6.80E-13	0.253932703	0.871	0.865	2.49E-08	B
TMA7	2.81E-81	-0.690348783	0.585	0.8	1.03E-76	B
HSP90AB1	1.83E-160	0.453956159	0.858	0.81	6.69E-156	CD4T
EIF4A1	5.78E-85	0.434797606	0.697	0.631	2.11E-80	CD4T
EZR	1.00E-18	0.356939422	0.526	0.495	3.68E-14	CD4T
CALM1	5.82E-120	0.345939726	0.914	0.913	2.13E-115	CD4T
HNRNPH1	1.99E-22	0.327495129	0.615	0.585	7.27E-18	CD4T
UBB	1.46E-55	0.313479219	0.758	0.734	5.35E-51	CD4T
MT-CO2	4.86E-88	0.3089001	0.998	0.995	1.78E-83	CD4T
RPS15A	4.99E-193	-0.265154655	0.966	0.973	1.83E-188	CD4T
S100A11	1.09E-52	-0.268939089	0.229	0.339	4.00E-48	CD4T
TMA7	0	-0.650830377	0.679	0.872	0	CD4T
UBB	4.96E-174	0.439335196	0.893	0.844	1.81E-169	CD8T
CALM1	2.54E-213	0.401543314	0.963	0.951	9.31E-209	CD8T
HSP90AB1	1.02E-182	0.399264575	0.912	0.885	3.75E-178	CD8T
EIF4A1	8.48E-90	0.357024201	0.76	0.688	3.11E-85	CD8T
EZR	4.69E-47	0.323653732	0.654	0.597	1.72E-42	CD8T
TMA7	0	-0.568076431	0.782	0.943	0	CD8T
RPS3	2.26E-07	-0.267467677	1	1	0.008271453	DC
TMA7	8.46E-16	-0.97083546	0.766	0.962	3.10E-11	DC
LMNA	2.21E-30	0.827099335	0.533	0.356	8.08E-26	gdT
HSP90AB1	2.12E-32	0.490151903	0.937	0.889	7.75E-28	gdT
EIF4A1	2.83E-17	0.383224597	0.843	0.778	1.04E-12	gdT
UBB	8.37E-21	0.352644339	0.956	0.948	3.06E-16	gdT
CALM1	1.30E-21	0.347910828	0.973	0.972	4.77E-17	gdT
EZR	6.19E-08	0.258447329	0.805	0.771	0.002267182	gdT
ATP5MG	3.91E-14	-0.256414135	0.821	0.898	1.43E-09	gdT
PFN1	1.31E-09	-0.268251868	0.871	0.937	4.81E-05	gdT
RAC2	1.25E-11	-0.312955641	0.519	0.642	4.58E-07	gdT
TMA7	5.10E-84	-0.702712927	0.737	0.939	1.87E-79	gdT
HSP90AB1	6.28E-08	0.335684502	0.958	0.949	0.00230029	MAIT
RPS16	1.23E-14	-0.276806518	0.988	1	4.50E-10	MAIT
TMA7	3.65E-29	-0.639325892	0.705	0.906	1.34E-24	MAIT
HSP90AB1	1.57E-23	0.572602164	0.66	0.638	5.74E-19	Mono_CD14
CALM1	2.40E-32	0.509200123	0.741	0.708	8.79E-28	Mono_CD14
UBB	2.40E-09	0.312830656	0.689	0.707	8.79E-05	Mono_CD14
GNB2	2.22E-25	-0.252939786	0.486	0.642	8.13E-21	Mono_CD14
COX4I1	1.58E-42	-0.281643843	0.881	0.947	5.77E-38	Mono_CD14
S100A11	1.27E-28	-0.283020982	0.721	0.877	4.66E-24	Mono_CD14
HSP90B1	1.71E-27	-0.292995943	0.703	0.833	6.25E-23	Mono_CD14
NAMPT	2.50E-29	-0.295865622	0.695	0.862	9.16E-25	Mono_CD14
RPS16	1.66E-50	-0.298710517	0.91	0.956	6.07E-46	Mono_CD14

RPL13A	1.92E-42	-0.324845051	0.825	0.927	7.04E-38	Mono_CD14
RPL23	2.91E-34	-0.335809076	0.683	0.839	1.06E-29	Mono_CD14
MIF	6.88E-36	-0.346298704	0.583	0.751	2.52E-31	Mono_CD14
S100A9	5.90E-46	-0.418282844	0.783	0.891	2.16E-41	Mono_CD14
CSTA	3.26E-74	-0.54328084	0.55	0.772	1.19E-69	Mono_CD14
TMA7	4.10E-197	-0.894289409	0.535	0.833	1.50E-192	Mono_CD14
TMA7	1.31E-28	-0.983904138	0.693	0.935	4.80E-24	Mono_CD16
HSP90AB1	5.20E-184	0.469828473	0.828	0.772	1.90E-179	NK
HNRNPH1	2.94E-24	0.36253915	0.532	0.518	1.08E-19	NK
CALM1	9.76E-179	0.351216111	0.927	0.927	3.57E-174	NK
LMNA	8.59E-41	0.343154517	0.505	0.455	3.14E-36	NK
DYNC1H1	3.86E-18	0.310286729	0.435	0.415	1.41E-13	NK
MT-CO2	6.64E-139	0.293152861	0.997	0.996	2.43E-134	NK
WIPF1	1.34E-86	-0.252985597	0.579	0.723	4.89E-82	NK
HNRNPA3	1.38E-106	-0.288782537	0.564	0.718	5.03E-102	NK
MIF	4.81E-107	-0.319405355	0.545	0.701	1.76E-102	NK
HSP90B1	1.13E-76	-0.340905264	0.712	0.807	4.12E-72	NK
TMA7	0	-0.488810548	0.695	0.881	0	NK
FCER1G	1.54E-292	-0.646224326	0.236	0.467	5.63E-288	NK
HSP90AB1	9.88E-64	0.349389967	0.614	0.564	3.62E-59	Other_Mono
CALM1	5.06E-58	0.3077343	0.681	0.64	1.85E-53	Other_Mono
S100A9	6.37E-111	-0.385917249	0.311	0.473	2.33E-106	Other_Mono
TMA7	3.15E-258	-0.720556012	0.279	0.502	1.15E-253	Other_Mono
HSP90AB1	7.59E-17	0.4613491	0.932	0.9	2.78E-12	Treg
CALM1	1.51E-17	0.436754432	0.981	0.953	5.51E-13	Treg
UBB	1.94E-07	0.331781214	0.879	0.832	0.00711723	Treg
TMA7	2.04E-41	-0.81096236	0.72	0.932	7.47E-37	Treg

Methods

- What time of the day was the blood sample obtained for cortisol? The authors mentioned same time of a day but did not specify when. Ideally you want all participants to get bloodwork in the morning, as 8-9am cortisol levels would be more meaningful.

Response: Thank you for the comment. All samples were collected between 7-9am, Beijing time, because all volunteers were empty stomached for at least 8 hours before sampling. We have added this information in the revised manuscript. Thank you.

Minor

1. Line 107-108 This suggests the active clearance of exogenous antigens from the body during the first three months after BA.2 BTI not sure if this is true regarding elevated monocyte %.

Response: Thank you for the comment. That was our hypothesis because it is well known that monocytes play important roles in antigen phagocytosis, clearance, and defense. We have adjusted the description in the manuscript.

2. I am just curious if you see any correlation between B cell activation/IFN related

signal/antigen processing from scRNA-seq (e.g. antigen processing module score, IRF7, ATF3 etc.) and neutralization levels to different subvariant. Would be interesting to make a correlation heatmap or something equivalent to link pseudovirus neutralization levels and B cell markers at individual participant levels.

Response: Thank you for the comment. We have supplemented the analysis of the correlation between the antigen presentation, interferon, and B cell activation module scores of 5 breakthrough infected individuals and the neutralizing activity of different virus strains. The results are shown in the figure below. Overall, antigen presentation, interferon, B cell activation, and BCR signaling are positively correlated with the neutralizing activity of various SARS-CoV-2 variants, while SARS-CoV is not. The JAK-STAT signaling pathway related to antiviral activity shows a negative correlation with SARS-CoV neutralizing activity. We have updated the manuscript, please have a look.

Figure R7. Correlation between antigen presentation, interferon and B cell activation module scoring and neutralizing activity to different virus strains.

3. For the discussion of cortisol, it is also associated with stress levels (for example <https://academic.oup.com/jcem/article/90/8/4579/3058896>).

Response: Thank you for the valuable comment. We have updated the main text accordingly. Cortisol plays a key role in regulating homeostatic stress responses, and hypocortisolism has substantial clinical overlap with COVID-19 symptoms. But cortisol may be higher during moderate-intensity stress (<https://academic.oup.com/jcem/article/90/8/4579/3058896>). Research showed that post-COVID-19 unemployment rate increases during the epidemic (doi: <https://doi.org/10.1101/2022.08.09.22278592>), and participants' concern about their own recovery after infection, and the stress of life and work may induce an increase in cortisol. In addition, a low level of cortisol was listed as an independent predictive biomarker of Long COVID after PT infection(<https://doi.org/10.1101/2022.08.09.22278592>), whether it is also applicable to the development of COVID-19 after Omicron BTI remains to be studied. They also discussed possible mechanisms, they proposed that this hypocortisolism was not associated with a significant perturbation in ACTH levels, suggesting an inappropriately blunted compensatory response by the hypothalamic-pituitary axis, and proposed the need for expanded investigation (doi: <https://doi.org/10.1101/2022.08.09.22278592>).

Reviewer #2 (Remarks to the Author):

Li et al. investigate proteomic and transcriptional landscape of blood samples after breakthrough infections (BTI) with the Omicron variant of COVID-19 with up to 6 month follow up. They measured clinical blood parameters for liver, renal and hematopoietic systems for 60 patients with BTI and 20 controls. They also perform virus neutralisation assays, blood proteomics and single cell transcriptomics on a subset of patients. The study is largely descriptive, with limited donors profiled in depth for example only 9 donors profiled with single cell transcriptomics. Although the authors discuss long COVID in text, a key piece of data missing is the symptomatic landscape of these patients with omicron BA.2 BTI. In addition, there have been many studies of long COVID/post-acute sequelae of COVID (PACS), albeit without the context of BTI infection, which the authors do not contextualise/contrast their findings to. Expanding on this, I have some specific comments below:

Response: Thank you for the valuable suggestion. We have updated the Discussion section, please refer to the revised manuscript, Thanks you.

1. The serology data is the most comprehensive in terms of patient numbers, are the authors able to quantify symptoms of long COVID in these patients (eg. from patient data or surveys) to further stratify those with BTI and long COVID symptoms vs those without?

Response: Thank you for the valuable suggestion. We have updated this information in the main text. We followed-up symptom information of the participants with phone call and clinical materials, collecting if they have long-term symptoms. Among the 60 participants who experienced mild-moderate BTI, four people had lung shadows visible on CT scanning after infection and were treated with antiviral + non-steroidal anti-inflammatory drugs. The others had no apparent lung abnormalities and were treated symptomatically according to their personal wishes. At three months after BTI, the abnormal lung CT manifestations of the four people disappeared. Only four people (4/60, 6.7%) (not limited to those with abnormal CT) had physical strength, lung function, and vestibular function abnormalities at three months after BTI, though, no obvious residual symptoms were left at 6 months post BTI. When comparing a large sample size with a small sample size, if the sample size difference is too large, the error on one side will be large and the conclusion will be less reliable. Thus, we neither emphasized symptoms nor further stratified those with BTI and long COVID symptoms vs those without.

2. Did the authors check for other infections/recent illness history?

Response: Thank you for the comment. We did not check other possible infections as no recent illness were claimed during sampling. This might be a limitation of our study. We have updated the limitation section.

3. Figure 2 shows increased neutralisation after break through infection – this is known in other vaccination settings albeit to an older set of variants, this study should be cited: <https://www.science.org/doi/10.1126/scitranslmed.abn8057>

Response: Thank you for the valuable suggestion. We have updated the citations.

4. When describing figure 3d and c, the authors refer to core hub proteins but it is unclear what this means.

Response: Thank you for the comment. The hub proteins here refer to proteins that participate in multiple biological processes simultaneously, and they are associated with multiple biological processes. Based on your suggestion, we have removed the description of hub proteins and switched to directly describing their regulatory processes. Thank you.

5. How exactly did the authors identify that there were no batch effects in their single cell data integration? Did they test or bench mark different batch correction methods?

Response: Thank you for your comment. Our single cell data analysis has included removal of batch effects from the Methods description. Specifically, Harmony R package was used to remove batch effects between individuals and groups [DOI: 10.1038/s41592-019-0619-0]. In the new data analysis, Seurat's built-in Anchor integration module was used to remove batch effects between samples in Canonical Correlation Analysis (CCA) [DOI: 10.1016/j.cell.2019.05.031]. Based on your suggestion, we have supplemented the cell clustering results before and after removing the batch effects. As shown in the figure below, before removing batch effects, different cell clusters exhibit uneven distribution between different samples or groups, indicating that they may be clustered together according to sample or inter group effects. On the contrary, after removing the batch effects, the distribution of different cell clusters between the samples and groups was more uniform, which means that the batch effects were removed at this time.

Figure R8. Batch Effects Removal for Single-cell Data Analysis. A: Unsupervised clustering of cells before and after batch effects removal. B: Cell cluster distribution of different individuals before and after batch effects removal. C: Cell cluster distribution of different groups before and after batch effects removal.

6. For single cell data overview, could the authors distinguish CD4 and CD8 T cells? I understand T cells are further subclustered in Figure 6, but this would be useful in the global analysis presented in Figure 4.

Response: Thank you for the valuable suggestion. We have subdivided the single cells in Figure 4 based on CD4 and CD8 subpopulations and conducted subsequent analysis based on this. Thank you.

Figure R9. Subdivided the CD4 and CD8 T cell subsets of Figure 4 based on the comments.
 A: The UMAP diagram displays cell types based on Azimuth multimodal algorithm and manual identification. B: The UMAP plot shows the distribution of cell types in the control and BA.2-BTI-6m groups. C: The bubble chart displays specific highly expressed markers of different cell types.

7. The authors assess differences in cell percentages, there are more sophisticated methods that could be used instead for example those summarised in: <https://www.biorxiv.org/content/10.1101/2023.02.24.529894v1.full> using a method such as Milo is independent of cell labels and may reveal hidden differences.

Response: Thank you for your valuable comment. We have applied the Milo algorithm to the reanalyzed single-cell data. As shown in the following Figure R10 D, E, and F, we still did not find any significant differences in the proportion of cell populations between the control group and the BA.2-BTI-6m group. We believe this is in line with the clinical results of our study. Another possible important and interesting reason might be that the long-term changes in PBMC induced by SARS-CoV-2 are not due to cell proportion but rather to transcriptional expression or epigenetic changes.

Figure R10. Identification of potential differences in cell populations based on Milo algorithm. A: The UMAP diagram displays cell types based on Azimuth multimodal algorithm and manual identification. B: The UMAP plot shows the distribution of cell types in the control and BA.2-BTI-6m groups. C: The bubble chart displays specific highly expressed genes of different cell types. D: The proportion of different types of cells in each sample. E: The UMAP graph shows the identification of potential differences in cell populations based on the Milo algorithm. F: The proportion difference and significance distribution of cell types identified by Milo algorithm.

8. For differential gene expression analysis, a more robust method is to perform pseudobulking per cell type/donor and differential expression using bulk tools such as edgeR.

Response: Thank you for your comment. We had tried pseudo differential expression analysis before, but the number of differentially expressed genes (DEGs) identified in our data was very small and manual validation results showed obvious false negatives, making it impossible to explain the results. Therefore, we believe that specific differential gene calculation methods may require more flexible and reasonable choices. We ultimately chose the more commonly used Findmarker() calculation module from Seurat, which is more in line with the difference analysis results verified manually and visually. Although conducting rank sum tests on single

cells may have potential drawbacks such as low significance values and susceptibility to individual sample influences (PMID: 34584091). When calculating DEG, in order to improve the accuracy of DEG screening as much as possible, we have raised the screening threshold. For example, setting $FDR < 0.01$ results in maintaining a lower P value (< 0.000001). In addition, DEG also needs to be expressed in at least 25% of cells in any group and detected in at least 20 cells (default thresholds: 10% and 3) to ensure that the gene is expressed to a certain extent in multiple samples and avoid extreme single sample expression. Similar DEG identification schemes have also been adopted in multiple recent literatures [PMID: 37776858, PMID: 37776858, PMID: 35507870].

For example, in terms of our data, we used pseudo differential expression and old Findmarker to calculate the DEGs and made a simple comparison. The results showed that pseudo differential expression analysis identified a total of 220 DEGs in 13 major cell types ($FDR < 0.05$ | $\log FC < 0.25$), and the number of DEGs in multiple cell types was less than 10 except for NK cells, $\gamma\delta$ T cells and Treg cells. On the contrary, Findmarker identified a total of 1554 DEGs ($FDR < 0.01$ | $\log FC < 0.25$) in different cell types, and the number of DEGs in different cell types remained mainly between 8 and 305. Among them, 310 DEGs showed greater changes ($|\log FC| > 0.5$), and for fine cell populations, the accumulation of genes with small changes may still exert biological functions.

Finally, Taking DNAJB1 and FOSB, which was significantly upregulated in B cells and $\gamma\delta$ T cells, respectively, and they were identified in the Findmarker calculation process but not found in pseudo differences. The manual visualization results indicate that they are clearly upregulated in the BA.2-BTI-6m group, both in the sample and between groups.

Figure R11. Visualization of differential genes missing from pseudo differential analysis. A-B: The violin diagrams showing the expression levels of DNAJB1 in B cells of different samples and groups. C-D: The violin diagrams showing the expression levels of FOSB in $\gamma\delta$ T cells of different samples and groups. Both differentially expressed genes were discovered by the Findmarker pipeline but were not identified by pseudo differential analysis.

Table R3. The differentially expressed genes identified by the pseudo differential expression analysis pipeline in the main cell cluster.

cell_type	gene	avg_logFC	p_val	p_val_adj	de_family	de_method	de_type
-----------	------	-----------	-------	-----------	-----------	-----------	---------

B	CTAG2	-3.885541313	1.07E-07	0.001724896	pseudobulk	edgeR	LRT
B	DDX43	2.123759355	6.60E-07	0.005315309	pseudobulk	edgeR	LRT
B	FOSB	0.880335329	7.03E-06	0.022668779	pseudobulk	edgeR	LRT
B	IGHV3-21	1.32240863	2.32E-05	0.041570308	pseudobulk	edgeR	LRT
B	IGHV4-61	3.177306946	1.49E-06	0.007982107	pseudobulk	edgeR	LRT
B	IGKV1-16	-1.207357613	1.49E-05	0.034306151	pseudobulk	edgeR	LRT
B	IGLV3-12	5.73451111	1.10E-05	0.029546017	pseudobulk	edgeR	LRT
B	RHOB	0.699242602	2.32E-05	0.041570308	pseudobulk	edgeR	LRT
B	S100B	-5.272553494	4.13E-06	0.016627412	pseudobulk	edgeR	LRT
CD4T	LERFS	-3.685381026	1.04E-07	0.00195273	pseudobulk	edgeR	LRT
DC	CXCL10	2.797737458	8.05E-11	8.06E-07	pseudobulk	edgeR	LRT
DC	HSPA1A	2.825834462	2.43E-08	6.09E-05	pseudobulk	edgeR	LRT
DC	IGLV2-14	4.543393768	1.90E-08	6.09E-05	pseudobulk	edgeR	LRT
DC	MT2A	1.983816437	1.23E-05	0.020456512	pseudobulk	edgeR	LRT
DC	PTGDS	-3.335211245	7.30E-09	3.66E-05	pseudobulk	edgeR	LRT
DC	TMEM176A	3.992417399	3.92E-08	7.85E-05	pseudobulk	edgeR	LRT
MAIT	TRBV5-1	3.179407981	8.73E-07	0.010778639	pseudobulk	edgeR	LRT
Mono_CD14	AC005392.2	-2.586590339	6.55E-06	0.016144402	pseudobulk	edgeR	LRT
Mono_CD14	CTAG2	-4.747112764	3.53E-07	0.001923887	pseudobulk	edgeR	LRT
Mono_CD14	FOLR3	-3.991773961	1.36E-06	0.005574227	pseudobulk	edgeR	LRT
Mono_CD14	HBB	-5.040541125	6.43E-06	0.016144402	pseudobulk	edgeR	LRT
Mono_CD14	HILPDA	-1.831836666	5.57E-08	0.000455114	pseudobulk	edgeR	LRT
Mono_CD14	IGHV3-23	5.08104113	6.91E-06	0.016144402	pseudobulk	edgeR	LRT
Mono_CD14	S100P	-3.204315272	2.90E-08	0.000455114	pseudobulk	edgeR	LRT
Mono_CD16	CLEC4F	-5.109820631	7.43E-06	0.042739688	pseudobulk	edgeR	LRT
Mono_CD16	PTGDS	-5.480783844	7.86E-13	9.05E-09	pseudobulk	edgeR	LRT
NK	AC007278.1	-1.470535138	0.00018799	0.03931452	pseudobulk	edgeR	LRT
NK	AC007952.4	-1.070229419	9.69E-08	0.000621226	pseudobulk	edgeR	LRT
NK	AC008840.1	-1.859594054	8.26E-05	0.02632291	pseudobulk	edgeR	LRT
NK	AC017104.1	-1.045549803	0.000161542	0.037000922	pseudobulk	edgeR	LRT
NK	AC018450.1	-1.473740579	0.000132483	0.034980164	pseudobulk	edgeR	LRT
NK	AC100830.1	-3.770942901	0.000246285	0.044023745	pseudobulk	edgeR	LRT
NK	AC103691.1	-2.752686645	1.56E-06	0.00203731	pseudobulk	edgeR	LRT
NK	AC124248.1	-1.905686998	9.24E-06	0.006879999	pseudobulk	edgeR	LRT
NK	AC138649.1	-1.667357716	0.000198924	0.040715841	pseudobulk	edgeR	LRT
NK	ACCS	1.245544761	1.25E-06	0.002019303	pseudobulk	edgeR	LRT
NK	ADAMTS10	-0.732483372	0.000203843	0.041283633	pseudobulk	edgeR	LRT
NK	AK5	-2.999918981	2.67E-05	0.014253285	pseudobulk	edgeR	LRT
NK	AKIRIN2	0.686268128	0.000106846	0.030231177	pseudobulk	edgeR	LRT
NK	AL353807.2	4.492884336	0.000123857	0.033563593	pseudobulk	edgeR	LRT
NK	AL358333.3	-1.209043608	0.000230003	0.043384931	pseudobulk	edgeR	LRT
NK	ALOX5AP	-0.848108431	1.02E-06	0.001953961	pseudobulk	edgeR	LRT
NK	AP3M2	0.754829637	0.000277645	0.04655308	pseudobulk	edgeR	LRT
NK	ARHGEF10	-2.529978373	3.58E-06	0.003826585	pseudobulk	edgeR	LRT

NK	ARL5B	-0.85323432	3.78E-05	0.016522485	pseudobulk	edgeR	LRT
NK	ASF1A	0.882403033	1.21E-05	0.007250211	pseudobulk	edgeR	LRT
NK	BAALC	-1.72040046	1.07E-05	0.007210361	pseudobulk	edgeR	LRT
NK	BAG3	1.060625516	6.78E-06	0.005927534	pseudobulk	edgeR	LRT
NK	BCL2L11	0.774823826	0.000263884	0.045739968	pseudobulk	edgeR	LRT
NK	BICDL1	1.492096348	3.54E-06	0.003826585	pseudobulk	edgeR	LRT
NK	BIRC2	-0.600363874	0.000145134	0.035799784	pseudobulk	edgeR	LRT
NK	CACNB1	0.866371796	0.000228241	0.043384931	pseudobulk	edgeR	LRT
NK	CAPN5	-1.179834934	1.13E-05	0.007210361	pseudobulk	edgeR	LRT
NK	CARD16	-0.733165119	0.000300516	0.048182668	pseudobulk	edgeR	LRT
NK	CD226	-0.816088731	0.000105133	0.030231177	pseudobulk	edgeR	LRT
NK	CD72	-1.031352038	2.84E-05	0.014394128	pseudobulk	edgeR	LRT
NK	CISH	1.04815674	0.000166063	0.037290385	pseudobulk	edgeR	LRT
NK	CKAP4	-1.071516336	0.00015877	0.037000922	pseudobulk	edgeR	LRT
NK	CLDND1	-0.697610276	7.48E-05	0.025253814	pseudobulk	edgeR	LRT
NK	CNFN	-1.814480859	1.26E-06	0.002019303	pseudobulk	edgeR	LRT
NK	COL13A1	-1.530764749	0.000219353	0.042629846	pseudobulk	edgeR	LRT
NK	COLQ	-1.509313943	6.82E-07	0.001619345	pseudobulk	edgeR	LRT
NK	CSF2	-1.978454084	5.18E-06	0.005248515	pseudobulk	edgeR	LRT
NK	DOHH	0.752531638	0.000282447	0.04655308	pseudobulk	edgeR	LRT
NK	DPY19L1	0.694276537	6.47E-05	0.023110865	pseudobulk	edgeR	LRT
NK	EIF5A	-0.517050727	0.000253193	0.044671951	pseudobulk	edgeR	LRT
NK	ERICH6-AS1	-0.80668833	3.02E-05	0.014887375	pseudobulk	edgeR	LRT
NK	FABP5	1.067347117	0.000233186	0.043558258	pseudobulk	edgeR	LRT
NK	FAM102A	0.96046218	9.65E-06	0.006879999	pseudobulk	edgeR	LRT
NK	FAM182B	-1.742866762	0.0001062	0.030231177	pseudobulk	edgeR	LRT
NK	FHL3	-0.867496563	3.74E-05	0.016522485	pseudobulk	edgeR	LRT
NK	FOSB	0.871583553	1.51E-06	0.00203731	pseudobulk	edgeR	LRT
NK	GLCE	1.477799636	4.22E-05	0.01725721	pseudobulk	edgeR	LRT
NK	GRASP	-0.753558658	4.81E-05	0.018493677	pseudobulk	edgeR	LRT
NK	GSC	2.505845024	0.000317882	0.049723999	pseudobulk	edgeR	LRT
NK	GSDMA	5.612328747	0.000289594	0.047039808	pseudobulk	edgeR	LRT
NK	GSN	-1.200400874	8.33E-05	0.02632291	pseudobulk	edgeR	LRT
NK	HBA2	-4.811428983	8.63E-05	0.026608481	pseudobulk	edgeR	LRT
NK	HBB	-4.102807031	4.01E-05	0.017151146	pseudobulk	edgeR	LRT
NK	HES4	-1.759108809	0.000246429	0.044023745	pseudobulk	edgeR	LRT
NK	HLA-DRB1	0.947609334	0.000310725	0.049002929	pseudobulk	edgeR	LRT
NK	HSPA6	-0.758033504	1.93E-05	0.010921483	pseudobulk	edgeR	LRT
NK	ICAM1	-1.036562863	4.18E-05	0.01725721	pseudobulk	edgeR	LRT
NK	IRGQ	-0.928305355	0.000152633	0.036708293	pseudobulk	edgeR	LRT
NK	ITGAD	4.234111104	4.49E-05	0.017708014	pseudobulk	edgeR	LRT
NK	ITM2C	-1.128584214	1.16E-05	0.007210361	pseudobulk	edgeR	LRT
NK	JPT1	1.057782081	0.00013728	0.034980164	pseudobulk	edgeR	LRT
NK	KCTD10	-0.849090637	1.10E-05	0.007210361	pseudobulk	edgeR	LRT

NK	KIF13A	-1.053978206	0.000186597	0.03931452	pseudobulk	edgeR	LRT
NK	LAIR2	-2.496222835	5.76E-06	0.005461917	pseudobulk	edgeR	LRT
NK	LINC00469	-1.888179529	0.000179927	0.038896681	pseudobulk	edgeR	LRT
NK	LZTFL1	-0.714306784	6.51E-05	0.023110865	pseudobulk	edgeR	LRT
NK	MAFB	1.153713302	5.96E-06	0.005461917	pseudobulk	edgeR	LRT
NK	MAN2A1	0.568792485	0.000290943	0.047039808	pseudobulk	edgeR	LRT
NK	MAS1	-3.147438981	8.71E-05	0.026608481	pseudobulk	edgeR	LRT
NK	MCTP1	-1.487242023	4.51E-05	0.017708014	pseudobulk	edgeR	LRT
NK	MLLT11	-0.822635273	7.68E-06	0.00615427	pseudobulk	edgeR	LRT
NK	MT1E	1.922756208	0.000138175	0.034980164	pseudobulk	edgeR	LRT
NK	MVB12B	0.916440049	0.000218384	0.042629846	pseudobulk	edgeR	LRT
NK	NANS	0.723371498	0.000281476	0.04655308	pseudobulk	edgeR	LRT
NK	NDRG1	1.045724574	7.57E-07	0.001619345	pseudobulk	edgeR	LRT
NK	NDST1	-2.352320832	0.000283093	0.04655308	pseudobulk	edgeR	LRT
NK	NFE2L2	-0.578032509	0.000247119	0.044023745	pseudobulk	edgeR	LRT
NK	NFKBID	-1.263260729	0.000136175	0.034980164	pseudobulk	edgeR	LRT
NK	OTUD1	1.18265707	0.000227047	0.043384931	pseudobulk	edgeR	LRT
NK	OTUD7A	-1.001345743	8.35E-05	0.02632291	pseudobulk	edgeR	LRT
NK	OTULIN	-0.838683578	1.05E-08	0.000116286	pseudobulk	edgeR	LRT
NK	P2RX5	-1.034929468	6.66E-07	0.001619345	pseudobulk	edgeR	LRT
NK	PADI4	-2.369195597	0.000307506	0.048896031	pseudobulk	edgeR	LRT
NK	PAFAH2	-0.631416061	0.00024701	0.044023745	pseudobulk	edgeR	LRT
NK	PCYOX1	1.026013178	0.000168621	0.037290385	pseudobulk	edgeR	LRT
NK	PDXDC1	-0.708025026	9.45E-06	0.006879999	pseudobulk	edgeR	LRT
NK	PDZD2	-1.759035649	0.000142356	0.035570606	pseudobulk	edgeR	LRT
NK	PLK3	0.782586652	0.000136386	0.034980164	pseudobulk	edgeR	LRT
NK	PRKAR1A	-0.588092322	0.000207038	0.041493851	pseudobulk	edgeR	LRT
NK	PTPRM	1.910261979	6.73E-05	0.023110865	pseudobulk	edgeR	LRT
NK	RARA-AS1	-1.118963713	0.000179844	0.038896681	pseudobulk	edgeR	LRT
NK	RBPMS2	-2.654887243	2.34E-06	0.002808328	pseudobulk	edgeR	LRT
NK	RHOBTB3	-0.807971885	6.49E-05	0.023110865	pseudobulk	edgeR	LRT
NK	RIN3	-0.590012743	9.47E-05	0.028473966	pseudobulk	edgeR	LRT
NK	S1PR5	-0.540378978	0.000166737	0.037290385	pseudobulk	edgeR	LRT
NK	SCRN1	-1.576945647	0.000149551	0.036422185	pseudobulk	edgeR	LRT
NK	SIGLEC7	-1.038289398	0.000192344	0.039792393	pseudobulk	edgeR	LRT
NK	SLC35G2	1.548699847	0.000255401	0.044671951	pseudobulk	edgeR	LRT
NK	SLC6A4	-1.702760918	3.54E-05	0.016202461	pseudobulk	edgeR	LRT
NK	SMIM3	-1.541860497	3.31E-05	0.015527239	pseudobulk	edgeR	LRT
NK	SPTLC3	-3.473382675	3.11E-05	0.014951448	pseudobulk	edgeR	LRT
NK	STARD3NL	-0.575428305	2.75E-05	0.01431436	pseudobulk	edgeR	LRT
NK	SYNJ2	-1.248128653	1.59E-06	0.00203731	pseudobulk	edgeR	LRT
NK	TLE1	-0.958151941	6.04E-05	0.022785675	pseudobulk	edgeR	LRT
NK	TMIGD2	-1.493541931	2.00E-05	0.011019433	pseudobulk	edgeR	LRT
NK	TNFSF14	-0.907429683	1.72E-07	0.000827758	pseudobulk	edgeR	LRT

NK	TOB2	0.695090154	0.000161509	0.037000922	pseudobulk	edgeR	LRT
NK	TRGV9	-0.920002804	2.64E-07	0.001016163	pseudobulk	edgeR	LRT
NK	TSPAN3	-0.603785504	0.00027603	0.04655308	pseudobulk	edgeR	LRT
NK	UEVLD	-0.719495648	0.000243881	0.044023745	pseudobulk	edgeR	LRT
NK	USP13	1.164053372	0.000122336	0.033563593	pseudobulk	edgeR	LRT
NK	YBX3	1.758669755	1.92E-05	0.010921483	pseudobulk	edgeR	LRT
NK	ZBTB38	0.665838674	0.000111232	0.031016023	pseudobulk	edgeR	LRT
NK	ZCCHC10	-0.686128781	7.37E-06	0.00615427	pseudobulk	edgeR	LRT
NK	ZMAT4	-1.983493205	1.21E-08	0.000116286	pseudobulk	edgeR	LRT
NK	ZNF260	-0.949589361	3.21E-07	0.001028007	pseudobulk	edgeR	LRT
NK	ZNF516	-1.593682342	6.67E-05	0.023110865	pseudobulk	edgeR	LRT
NK	ZNF529	-0.65774406	7.78E-05	0.025800292	pseudobulk	edgeR	LRT
NK	ZNF595	-1.097698588	0.000216791	0.042629846	pseudobulk	edgeR	LRT
NK	ZNF618	-1.473923987	0.000103021	0.030231177	pseudobulk	edgeR	LRT
NK	ZNF674-AS1	-0.96398944	0.000182866	0.03909279	pseudobulk	edgeR	LRT
NK	ZNF710	-0.701222277	0.000154778	0.036764455	pseudobulk	edgeR	LRT
NK	ZRANB2	-0.597654933	0.000273682	0.04655308	pseudobulk	edgeR	LRT
Other_Mono	ARHGEF10	-3.584377531	6.97E-06	0.021054251	pseudobulk	edgeR	LRT
Other_Mono	CCNE2	2.046511355	1.14E-08	0.000206464	pseudobulk	edgeR	LRT
Other_Mono	EFNB2	1.488923863	2.30E-05	0.041699269	pseudobulk	edgeR	LRT
Other_Mono	EOMES	1.179975843	6.23E-06	0.021054251	pseudobulk	edgeR	LRT
Other_Mono	FSCN1	1.583848611	3.99E-08	0.000361351	pseudobulk	edgeR	LRT
Other_Mono	KLRC1	-1.458645562	9.75E-07	0.004417827	pseudobulk	edgeR	LRT
Other_Mono	PRDX1	0.810305757	1.18E-05	0.030416165	pseudobulk	edgeR	LRT
Other_Mono	VASN	1.917942061	1.62E-05	0.032624978	pseudobulk	edgeR	LRT
Other_Mono	XRRA1	1.738099349	1.40E-05	0.031816439	pseudobulk	edgeR	LRT
Other_Mono	ZMAT4	-2.157631763	3.46E-07	0.00208997	pseudobulk	edgeR	LRT
Platelet	CD74	-2.585894197	1.43E-06	0.011320063	pseudobulk	edgeR	LRT
Platelet	HBG2	-3.383753741	1.05E-05	0.041601596	pseudobulk	edgeR	LRT
Treg	AC116366.1	2.014807514	3.13E-07	0.001332136	pseudobulk	edgeR	LRT
Treg	BIN1	-0.86510005	3.07E-05	0.026138585	pseudobulk	edgeR	LRT
Treg	CCR6	-1.0225306	3.46E-05	0.027586618	pseudobulk	edgeR	LRT
Treg	CD83	1.015812785	2.75E-06	0.00701857	pseudobulk	edgeR	LRT
Treg	FABP5	1.050215394	7.39E-05	0.047161445	pseudobulk	edgeR	LRT
Treg	FGL2	-2.735616096	2.15E-05	0.022855812	pseudobulk	edgeR	LRT
Treg	FOSB	0.898808484	4.18E-06	0.007619712	pseudobulk	edgeR	LRT
Treg	GNLY	1.25618234	4.87E-05	0.034617598	pseudobulk	edgeR	LRT
Treg	IFIT2	1.301234053	1.93E-05	0.022340683	pseudobulk	edgeR	LRT
Treg	MAP1A	2.884797402	1.77E-05	0.022340683	pseudobulk	edgeR	LRT
Treg	NR4A1	1.051661441	2.54E-05	0.023121077	pseudobulk	edgeR	LRT
Treg	OAT	1.038197816	4.07E-06	0.007619712	pseudobulk	edgeR	LRT
Treg	RHOB	1.711417313	6.91E-10	8.82E-06	pseudobulk	edgeR	LRT
Treg	RPS26	-1.131688881	6.90E-07	0.002201065	pseudobulk	edgeR	LRT
Treg	SMDT1	-0.802039677	8.23E-05	0.049978773	pseudobulk	edgeR	LRT

Treg	TMA7	-0.814950588	7.93E-06	0.012637899	pseudobulk	edgeR	LRT
Treg	TRBV12-3	-1.823119859	5.89E-05	0.039525303	pseudobulk	edgeR	LRT
Treg	TRBV14	1.845180636	2.44E-05	0.023121077	pseudobulk	edgeR	LRT
Treg	TRBV6-1	-2.14654173	1.37E-07	0.000871704	pseudobulk	edgeR	LRT
Treg	UBE2D1	1.425649679	1.27E-05	0.017981176	pseudobulk	edgeR	LRT
Treg	ZNF296	1.764065931	4.88E-05	0.034617598	pseudobulk	edgeR	LRT
gdT	AC044849.1	1.109633798	8.25E-05	0.04655178	pseudobulk	edgeR	LRT
gdT	AC116366.1	1.127133408	2.49E-05	0.026092948	pseudobulk	edgeR	LRT
gdT	AC136475.3	1.191618536	0.000103094	0.04655178	pseudobulk	edgeR	LRT
gdT	AL590550.1	4.761844857	2.61E-05	0.026092948	pseudobulk	edgeR	LRT
gdT	AMN1	-2.47328413	6.87E-05	0.043859398	pseudobulk	edgeR	LRT
gdT	BIN1	-0.751894115	9.60E-05	0.04655178	pseudobulk	edgeR	LRT
gdT	BZW2	1.118737673	8.83E-05	0.04655178	pseudobulk	edgeR	LRT
gdT	CCNE2	2.430763181	3.32E-05	0.029065403	pseudobulk	edgeR	LRT
gdT	COL18A1	2.642031984	3.32E-05	0.029065403	pseudobulk	edgeR	LRT
gdT	ETV7	3.603855753	4.41E-05	0.036311029	pseudobulk	edgeR	LRT
gdT	HSD17B11	-0.819107995	0.000102012	0.04655178	pseudobulk	edgeR	LRT
gdT	HSPA2	1.183170087	8.18E-05	0.04655178	pseudobulk	edgeR	LRT
gdT	HSPH1	0.833498147	9.87E-05	0.04655178	pseudobulk	edgeR	LRT
gdT	ING1	1.116651513	2.09E-06	0.004874634	pseudobulk	edgeR	LRT
gdT	JPT1	1.139858091	4.39E-06	0.007967503	pseudobulk	edgeR	LRT
gdT	KLHL15	1.587476144	4.89E-05	0.038030696	pseudobulk	edgeR	LRT
gdT	KLRC3	2.071021792	6.83E-05	0.043859398	pseudobulk	edgeR	LRT
gdT	LAIR2	-2.908050141	6.00E-06	0.009327946	pseudobulk	edgeR	LRT
gdT	LMNA	1.161096264	3.90E-08	0.000545679	pseudobulk	edgeR	LRT
gdT	PGGHG	1.111600536	0.000109597	0.047414074	pseudobulk	edgeR	LRT
gdT	RASD1	1.236359634	5.21E-07	0.003100424	pseudobulk	edgeR	LRT
gdT	RCOR1	1.015460312	1.22E-05	0.017022525	pseudobulk	edgeR	LRT
gdT	SLC12A7	1.55625057	9.57E-05	0.04655178	pseudobulk	edgeR	LRT
gdT	SLC16A1	1.035165573	6.89E-05	0.043859398	pseudobulk	edgeR	LRT
gdT	SLC25A19	1.664321721	2.49E-05	0.026092948	pseudobulk	edgeR	LRT
gdT	TNFSF9	0.911941738	0.000111778	0.047414074	pseudobulk	edgeR	LRT
gdT	TOB2	1.056138881	6.64E-07	0.003100424	pseudobulk	edgeR	LRT
gdT	TRGV2	-1.396151665	2.00E-06	0.004874634	pseudobulk	edgeR	LRT
gdT	XCL1	-1.477107825	0.000100434	0.04655178	pseudobulk	edgeR	LRT
gdT	YBX3	1.085695432	1.64E-05	0.020849956	pseudobulk	edgeR	LRT
gdT	Z93241.1	1.144740748	5.65E-05	0.041653832	pseudobulk	edgeR	LRT
gdT	ZNF296	1.157380973	4.55E-06	0.007967503	pseudobulk	edgeR	LRT
gdT	ZNF683	-1.629995013	2.09E-06	0.004874634	pseudobulk	edgeR	LRT
pDC	CXCL10	2.704854609	2.44E-06	0.00643215	pseudobulk	edgeR	LRT
pDC	IFI30	-3.652904407	3.93E-08	0.00013825	pseudobulk	edgeR	LRT
pDC	IGHA1	-4.198113503	5.39E-09	5.69E-05	pseudobulk	edgeR	LRT
pDC	JCHAIN	-1.973522868	6.04E-06	0.012749439	pseudobulk	edgeR	LRT
pDC	RASD1	-2.644561787	2.04E-08	0.000107444	pseudobulk	edgeR	LRT

9. For the T cell annotations in Figure 6 the marker genes are not so distinct, could the authors display additional marker genes especially for NKT and CD8 subsets with one defining marker gene. It could also be good to compare the authors annotations to existing annotation from PBMC single cell data, for example using tools like Azimuth, CellTypist or integrating with existing PBMC/covid atlases.

Response: Thank you for your professional review. We have used the Azimuth algorithm to map our data to the reference cells of PBMC. Based on the cell types predicted by the Azimuth algorithm and combined with the top 50 highly expressed genes of each cluster, we have comprehensively determined the final cell type as shown in the figure below. Thank you again for your suggestion.

Figure R9. Subdivided the CD4 and CD8 T cell subsets of Figure 4 based on the comments.

10. How did they calculate scores for antigen presentation, protein processing and cytotoxicity in the single cell data?

Response: Thank you for your comment. We use the `AddModuleScore()` function of Seurat software to calculate the activity score for a specific gene set [PMID: 34062119, PMID: 27124452]. The calculation process of the `AddModuleScore()` function is as follows: Calculate the average expression levels of certain gene set on a single cell level, subtracted by the aggregated expression of control gene sets. All analyzed features were bound based on average expression, and the control features were randomly selected from each bin. The gene set was sourced from the `Msigdb` R package (KEGG-ANTIGEN-PROCESSING-AND-

PRESENTATION: hsa04520 and GOBP-PROTEIN_PROCESS: GO: 0016485) T cell toxicity activity was defined by the following gene sets: PRF1, IFNG, GNLY, NKG7, GZMB, GZMA, GZMH, KLRK1, KLRB1, KLRD1, CTSW, and CST7. Due to the word limit in the main text, please refer to the supplementary methods for this part of the methodology. Thank you.

11. Without increasing the number of patients in the single cell data, perhaps the authors could integrate their data with other single cell sequencing data from COVID patients followed longitudinally? For example <https://insight.jci.org/articles/view/165299>

Response: Thank you for the suggestion. The manuscript aimed to evaluate the immune recovery of patients with BA.2 BTI for 6 months, and the environment and baseline of the samples we used for sequencing were relatively matched. Considering that introducing other infection data can lead to many confounding factors, including sequencing batches, differences in SARS-CoV-2 strains, sampling, ethnicity, lifestyle habits, etc., we do not have confidence in effectively remove all the impact of these factors on the manuscript. Therefore, based on the valuable suggestions above, we have made every effort to conduct quality control and analysis of single cell data, which we believe greatly improves the results of the data without integrating other complex data. Thank you.

12. For the TCR and BCR repertoire – do they have this for 3 months post BTI? If so, could they incorporate it into the analysis.

Response: Thank you for the comment. In this work, our goal was to study the potential impacts on recovery of COVID, TCR and BCR repertoire analysis was more to supplement T cell response and B cell response status. We wanted to use sequencing as a tool to dig for what we did not find through routine blood tests and works for our scientific objectives. Therefore, we chose the samples on the 6-month time point for scRNA-Seq and we do not have TCR/BCR repertoire data for 3-month post BTI.

13. Again for the TCR and BCR data – did the authors identify any shared clones between individuals? For example, were there more shared clones in BTI individuals than controls?

Response: Thank you for your valuable comment. We have supplemented the analysis of TCR or BCR intersections in the BA.2-BTI-6m and Control groups. As shown in the figure (A, B, C), the moderately amplified TCR in the BA.2-BTI-6m group was 2.637% higher than that in the control group, while the difference in other types of TCR was about 1%. As expected, there was significant heterogeneity of TCR among different individuals, with few intersections being only 1 or 2. The results showed that a total of 6 TCR intersections (8 TCRs) were identified between the control group pairs, and a total of 6 TCR intersections (9 TCRs) were identified between the BA.2-BTI-6m group pairs. Despite limited sample size, preliminary results suggest that individuals with BA.2-BTI-6m may share no more TCR clones than the control group.

In addition, A slightly higher proportion of amplified BCR was observed in BA.2-BTI-6m compared to the control group (3.258% vs 2.336%) and we did not detect BCR intersections between individuals (D, E, F), mainly due to the high heterogeneity of the immune repertoire between individuals and the limited number of B cells.

Figure R12. TCR or BCR shared by different samples. A: The proportion of TCR with different frequencies in the control group and BA.2-BTI-6m group. B: Intersection of TCR in control group samples. C: Intersection of TCR of samples from BA.2-BTI-6m group. D: The proportion of BCR with different frequencies in the control group and BA.2-BTI-6m group. E: Intersection of BCR samples in the control group. F: Intersection of BCR of samples in the BA.2-BTI-6m group.

Minor comments:

1. In the abstract the abbreviation PLT is used, please define this in the abstract.

Response: Thank you for the valuable comment. We have updated the abstract accordingly.

2. Please add the numbers of patients for each assay/data type in text and figure legends, this information was difficult to find.

Response: Thank you for the valuable suggestion. We have supplemented a table (Table S7) with numbers of patients for each data type.

3. In the text relating to Figure 6, the authors state that Memory T cells levels undergo intense and lasting dynamic shift. I believe the authors here are referring to differentially expressed

genes rather than T cell abundance (as the term level suggests). Regardless, I would say that this is an overstatement of the data.

Response: Thank you for the valuable comment. We have adjusted the tune in the main text.

Reviewer #3 (Remarks to the Author):

In the manuscript by Li et al. entitled Long-term effects of Omicron BA.2 breakthrough infection on immunity-metabolism balance: a 6-month prospective study the authors recruited individuals who received a vaccine against COVID-19 (2x Double Ad5-vectored COVID-19 vaccine; Convidecia, CanSino Bio, 398 Co., Ltd.) and developed a mild breakthrough infection (BTI) ~10 months after the last shot (n=60), and compared them to individuals who received the same vaccine but did not develop a BTI (n=20). Blood from individuals from the BTI group was collected 3 months and 6 months post infection; which corresponds to ~13 months and 16 months after the second vaccine shot, respectively. Blood from the control group was collected ~12 months after the second vaccine shot. The authors applied mass spectrometry-based proteomics of serum, scRNA-seq of PBMCs, and analysed clinical variables for a comparative analysis. The authors found differences including in coagulation, metabolism and anti-pathogen immunity.

Although the manuscript is methodically solid and well-written, the study allows limited conclusions because: i) the study is limited to relatively few patients and some experiments rely on few samples (e.g. proteomics of 18 samples; scRNA-seq of 9 samples); ii) the findings rely heavily on bioinformatical analyses with little biological validation; iii) no outcome data or information on development of long COVID is contained. The manuscript may therefore be better suited for lower-tiered journals of the Nature family such as Scientific Reports.

Comments to the authors

- Please discuss the following limitations: i) the study is limited to relatively few patients and some experiments rely on few samples (e.g. proteomics of 18 samples; scRNA-seq of 9 samples); ii) the findings rely heavily on bioinformatical analyses with little biological validation; iii) no outcome data or information on development of long COVID is contained.

Response: Thank you for the comments. i) The sample sizes of scRNA-seq and proteomics are not quite large but statistically rational. The limited sample size is a limitation of this study, and we have mentioned it in the Discussion section. It is indeed not easy to collect samples from vertical queues, which was something we have tried our best to obtain, especially under China's strict zero-COVID management in the past. We still conducted detailed data mining and analysis combining these sequencing data and clinical data to explore the recovery of the body up to 6-month after BTI. Additionally, there are evidence on very good quality scientific work with scRNA-seq of less than 5 samples (e.g. Zhu, Linnan, et al. Immunity 53.3 (2020): 685-696.). The proteomic analysis was seen as auxiliary for scRNA-seq and we had doubled the size to reversely validate the transcriptomic level changes at protein level. Importantly, after valuable suggestions from the reviewers, we believe that the manuscript has greatly improved

and hope to meet your requirements. Thank you again for your suggestion.

ii) Biological functions were performed usually at protein level, we had performed proteomics analysis and later validated the hypothesis in sera at clinical testable level. We have supplied information that has never been shown anywhere else and with mechanistic indications, which we believe could contribute to the understanding of the long-term effects of COVID.

iii) The definition of long COVID, “Some people who have been infected with the virus that causes COVID-19 can experience long-term effects from their infection, known as Long COVID or Post-COVID Conditions (PCC). Long COVID is broadly defined as signs, symptoms, and conditions that continue or develop after acute COVID-19 infection. This definition of Long COVID was developed by the Department of Health and Human Services (HHS) in collaboration with CDC and other partners.”(<https://www.cdc.gov/coronavirus/2019-ncov/long-term-effects/index.html>). Long COVID is not limited to the persistence of symptoms. Because, in this cohort, participants did not have residual symptoms at 6-month post BTI, we did not emphasize clinical symptoms or describe outcomes. Our topic is more on undergoing long-term effects of mild SARS-CoV-2 BA.2 BTI. The long-term effects are likely related to post COVID condition.

- Please provide a table that compares the baseline characteristics of the control and BTI groups.

Response: Thank you for the valuable comment. We have provided baseline characteristics in the revised Table S1, please refer to the modified file Table S1-7.

- Protein Abbreviations should be explained either in the text or in a table.

Response: Thank you for the comment. We have supplemented a table (Table S8) for protein abbreviations.

- The volcano plot in Figure 3b appears to only show nominal p-value. Please also report results adjusted for multiple testing.

Response: Thank you for your comment. Our screening criteria for identifying differential proteins in proteomics analysis are $pvalue < 0.05$, rather than overly strict FDR adjustments. The FDR adjustment may mainly be used for situations with a large number of genes, mainly to avoid low error multiplied by a large base, which can also accumulate a large number of false positives. Based on our understanding, for biological research designs with limited samples, FDR may be too strict to detect differentially expressed proteins. Therefore, it is necessary to choose pvalue or lower the FDR threshold based on the actual situation. In addition, the total number of quality-controlled proteins in our analysis was only 1646, rather than tens of thousands of genes, so there was no need for strict FDR adjustment, which may filter out many proteins and cause false negatives. The most direct evidence comes from the expression heat map, which showed that overall, there were significant differences in the 202 differentially expressed genes between the control group and the BA.2-BTI-6m group. Thank you again for your suggestion.

- Please explain the rationale for looking at PBMCs.

Response: Thank you for the comment. PBMCs are a set of cells, which are relatively reachable for biological research and can largely reflect the immune recovery of the patient's body after

infection, especially the evaluation of the patient's humoral and cellular immunity. This is one of the most common and efficient data for evaluating changes in immune cells in patients, compared to other sample types, as it is almost non-invasive and cost-effective. Please also check the updated manuscript. Thank you again for your suggestion.

- The finding of liver impairment is interesting and has already been found in the acute setting based on proteomics data (DOI:10.1038/s41467-021-23494-1) as well as circulating microRNA data (miR-192: DOI:10.1016/j.trsl.2021.05.004 ; miR-122 DOI:10.1093/cvr/cvab338) and should be discussed.

Response: Thank you for the comment. There are indeed research reports showing liver damage caused by COVID-19 infection. Most of the previous studies emphasized liver damage caused by moderate to severe infection. Here, we mainly update the in-depth assessment of the recovery situation within half a year after mild-moderate breakthrough infection with Omicron sub-variants. We have carefully read the three articles you mentioned. The first article (DOI:10.1038/s41467-021-23494-1) mainly focuses on hospitalized patients (general ward patients and ICU patients), and evaluates RNAemia and biomarkers related to severe disease and death. We have updated the Discussion section to integrate this literature. The latter two studies that mainly focus on microRNA and are not very relevant to our research content, so we did not discuss further in comparison.

- The authors should also discuss the role of RNAemia in the context of long COVID.

Response: Thank you for the comment. RNAemia plays a crucial role in the occurrence and development of long COVID. But about 95% of our research subjects have mild symptoms and RNAemia has not been found when we analyzed serum proteomics. Thus, as mentioned above, we only made a brief discussion.

Reviewers' Comments:

Reviewer #1:

Remarks to the Author:

I think the authors have addressed most of my questions and the quality of the manuscript has improved. Two minor suggestions

- For organ/tissue-specific GSEA on new Figure S4, please specify in the figure legend which database you used for reference.

- Line 275-279 and Fig 7g, given the small sample size (n=5 right?), some of these positive correlations did not reach statistical significance, would specify in the text that it showed a trend of positive correlation.

Reviewer #2:

Remarks to the Author:

Overall the authors have satisfied my comments, however I would like some further clarification on the point about DGE analysis. The authors performed pseudobulk + DGE analysis however they claim the results showed "obvious false negatives" based on "manual validation". Could the authors please clarify what "manual validation" means, and give examples? Furthermore I would encourage authors to show that any conclusions from DGE analysis using FindMarkers function in Seurat are not donor-driven, as validating this on the single cell level can lead to donor-specific results in cases where any particular cell type is over-represented by specific donors, or that cells from a specific donor have particularly high expression of a given gene. Again, could the authors also please clarify what they mean by "results were verified manually and visually when they say "We ultimately chose the more commonly used Findmarker() calculation module from Seurat, which is more in line with the difference analysis results verified manually and visually". Thanks in advance.

Reviewer #3:

Remarks to the Author:

As pointed out previously, the study is largely descriptive, with limited n-numbers in the -omics analysis, e.g. scRNA-seq of 9 samples and proteomics of 18 samples; the presented results are not adjusted for multiple testing. The authors response was as follows:

Response: ..Based on our understanding, for biological research designs with limited samples, FDR may be too strict to detect differentially expressed proteins. Therefore, it is necessary to choose p-value or lower the FDR threshold based on the actual situation. In addition, the total number of quality-controlled proteins in our analysis was only 1646, rather than tens of thousands of genes, so there was no need for strict FDR adjustment, which may filter out many proteins and cause false negatives. The most direct evidence comes from the expression heat map, which showed that overall, there were significant differences in the 202 differentially expressed genes between the control group and the BA.2-BTI-6m group. Thank you again for your suggestion.

I disagree that FDR adjustment is not necessary when there are "only" 1,646 proteins. When the authors base their findings on the nominal p-values, there is a high risk of false positives.

Reviewer 1:

1. For organ/tissue-specific GSEA on new Figure S4, please specify in the figure legend which database you used for reference.

Response: Thank you for your excellent suggestion. We have added the following information in the figure legend “The list of tissue-specific proteins used for GSEA analysis in the heart, liver, lungs, and pancreas was derived from the study by Jiang et al. (Cell. 2020, DOI: 10.1016/j.cell.2020.08.036). This study constructed a quantitative map of the proteome of different tissues in the human body.” Thank you again for your suggestion.

2. Line 275-279 and Fig 7g, given the small sample size (n=5 right?), some of these positive correlations did not reach statistical significance, would specify in the text that it showed a trend of positive correlation.

Response: Thank you for your valuable suggestion. We have made revisions to the manuscript (line 398-401, page 11), as “Next, we explored the correlation between neutralizing antibody levels and key signals in BA.2-BTI-6m subjects. Overall, in addition to the JAK-STAT signaling pathway, antigen presentation, interferon, B cell activation and BCR signaling were positively correlated with the neutralizing activity of various SARS-CoV-2 variants (Fig.7g). However, due to the small sample size, these positively correlated trends are not significant and therefore worth of further validation.” Thank you again for your suggestion.

Reviewer 2:

1. Overall the authors have satisfied my comments, however I would like some further clarification on the point about DGE analysis. The authors performed pseudobulk + DGE analysis however they claim the results showed "obvious false negatives" based on "manual validation". Could the authors please clarify what "manual validation" means, and give examples? Furthermore I would encourage authors to show that any conclusions from DGE analysis using FindMarkers function in Seurat are not donor-driven, as validating this on the single cell level can lead to donor-specific results in cases where any particular cell type is over-represented by specific donors, or that cells from a specific donor have particularly high expression of a given gene. Again, could the authors also please clarify what they mean by "results were verified manually and visually when they say "We ultimately chose the more commonly used Findmarker() calculation module from Seurat, which is more in line with the difference analysis results verified manually and visually". Thanks in advance.

Response: Thank you for your professional comments. The current options for analysis of single-cell differentially expressed genes (DEGs) are generally Findmarker() or pseudo differential analysis. We also believe that pseudo differential analysis performs well in many datasets, especially when there are large-size samples. However, in this manuscript, we attempted pseudo differential analysis but could hardly identify DEGs. We realized that pseudo differential analysis may not be able to explain the serum-level differences we had observed. Therefore, we chose Seurat's Findmarker() function to recalculate DEGs, and set the cutoff of differentially expressed genes to at least 25% (default parameter 10%) of cells in the control group or BA.2-BTI-6m group, with an $FDR < 0.01$, to reduce the impact of individual donors and false positives.

For manual validation, our previous description meant visualizing gene expression levels to see if there were different expression trends between groups and samples, which were true. For example, DEGs DNAJB1 of B cells and FOSB of $\gamma\delta$ T cells were calculated using the Findmarker(), but pseudo differential analysis did not identify them (Reply Figure A, C). The visualization of real expression levels indicates that although there are individual differences, the expression levels of these two genes in the BA.2-

BTI-6m group samples are higher than those in the control group samples (Reply Figure A, C). The visualization of expression levels between groups is more pronounced (Reply Figure B, D).

In summary, pseudo differential analysis may not be applicable to our small-size sample dataset, so we had to use Findmarker() to calculate DEGs, and to reduce false positives by setting strict cutoff values. Thank you again for your professional comments. Thank you.

Reply Figure. Visualization of differentially expressed genes missing from pseudo differential analysis.

A-B: The violin diagram shows the expression level of DNAJB1 in B cells of different samples and groups. C-D: The violin diagram shows the expression level of FOSB in $\gamma\delta$ T cells of different samples and groups. Both differentially expressed genes were discovered by the Findmarker() pipeline but were not able to be identified by pseudo differential analysis.

Reviewer3:

As pointed out previously, the study is largely descriptive, with limited n-numbers in the -omics analysis, e.g. scRNA-seq of 9 samples and proteomics of 18 samples; the presented results are not adjusted for multiple testing. The authors response was as follows:

Response: ..Based on our understanding, for biological research designs with limited samples, FDR may be too strict to detect differentially expressed proteins. Therefore, it is necessary to choose p-value or lower the FDR threshold based on the actual situation. In addition, the total number of quality-controlled proteins in our analysis was only 1646, rather than tens of thousands of genes, so there was no need for strict FDR adjustment, which may filter out many proteins and cause false negatives. The most direct evidence comes from the expression heat map, which showed that overall, there were significant differences in the 202 differentially expressed genes between the control group and the BA.2-BTI-6m group. Thank you again for your suggestion.

I disagree that FDR adjustment is not necessary when there are “only” 1,646 proteins. When the authors base their findings on the nominal p-values, there is a high risk of false positives.

Response: Thank you for your valuable comment. The goal of this study is to fill the gap in understanding the long-term effects of BA.2 breakthrough infection. Besides single-cell sequencing and proteomic analysis, for experimental validation, we have thoroughly tested the clinical indicators of patients and conducted neutralizing antibody evaluation targeting different variants to confirm the long-term effects of BA.2 breakthrough infection. Further in-depth experimental verification is indeed difficult to conduct due to the difficulty in tracking and sampling patients involved. We also recognized this limitation and discussed it in the discussion.

For the adjustment issue of FDR, we have placed all the raw protein expression matrix, fold of change, p-value, and q-value in Table S2. As we have observed and consulted with other data analysts, adjusting the FDR of bulk data in small samples can lead to an increase in false negatives. For example, in Table S2, there are 202 proteins with p-value<0.05, while there are only 24 proteins with q-value<0.05. However, we

found nearly 50 proteins with $p\text{-value} < 0.01$ but $q\text{-value} > 0.05$, and simply categorizing them as having no significant difference may be arbitrary and unreasonable. Moreover, from the original expression levels or heatmaps of these proteins, there is still a clear trend of differences (revised Figure 3). Thus, based on your suggestions and those of other data analysts, in order to minimize false positives, we used $q\text{-value}$ to re-calculate significance, and set the cutoff for differential proteins significance to $p\text{-value} < 0.05$ and $q\text{-value} < 0.25$, which is also the default FDR values for many algorithms. In the end, we obtained 80 differentially expressed proteins for subsequent analysis and updated the results related to proteomics (revised Figure 3). Thank you again for your suggestions.

Reviewers' Comments:

Reviewer #2:

Remarks to the Author:

Thank you for the considered answer, the authors have now satisfied all my queries, and I recommend the manuscript for acceptance.